# Gender roles in ruminant disease management in Uganda: Implications for the control of peste des petits ruminants and Rift Valley fever

Jane Namatovu[1,7]*, Peter Lule[1,2], Marsy Asindu[1,3], Zoë A. Campbell[7],
Dan Tumusiime[1,5], Henry Kiara[4], Bernard Bett[4], Kristina Roesel[4], Emily Ouma[6]

**1** Health, International Livestock Research Institute, Kampala, Uganda, **2** Department of Agricultural Economics, University of Nairobi, Nairobi, Kenya, **3** Department of Food Economics and Food Policy, Christian-Albrechts-University, Kiel, Germany, **4** Health, International Livestock Research Institute, Nairobi, Kenya, **5** Institute for Parasitology and Tropical Veterinary Medicine, Freie Universität Berlin, Berlin, Germany, **6** People, Policies and Institutions, International Livestock Research Institute, Kampala, Uganda, **7** People, Policies and Institutions Program, International Livestock Research Institute, Nairobi, Kenya

* J.Namatovu@cgiar.org

## Abstract

There is a distinct division of tasks and roles between men and women in livestock management in the different ruminant production systems in Uganda. Division of roles can influence disease control and prevention. This qualitative study asks what men and women do to prevent or control diseases that affect them and their livestock and what factors influence the choice of disease control measures taken. Discussants represented three production systems (pastoral, agro-pastoral, and mixed crop livestock), selected for the high prevalence of two livestock diseases (peste des petits ruminants and Rift Valley fever). Sex-disaggregated focus group discussions with livestock keepers and key informant interviews with veterinarians and other experts were conducted in six districts in the western, northeastern, and eastern regions of Uganda. Gendered livestock management roles strategically positioned men, women, girls, and boys to observe different clinical manifestations of disease. Livestock keepers mostly reported within-farm prevention and control methods, for which they presumably had more control than between-farm or community-level methods. While livestock keepers embraced disease control options such as the use of drugs, spraying acaricides, and the use of traditional herbs, many had concerns and misconceptions about vaccination as a preventive measure. Although women had fewer concerns about vaccine side effects, they still faced constraints, such as mistrust of animal health workers, limited decision-making powers, domestic workload, and inability to access vaccination points. The study findings can guide appropriate, gender-responsive interventions tailored by production system for controlling ruminant diseases in Uganda.

**Data availability statement:** All relevant data files and related metadata underlying the reported findings are openly available from the Harvard Dataverse at https://doi.org/10.7910/DVN/UM9HIN.

**Funding:** This work was supported by the German Federal Ministry of Economic Cooperation and Development (BMZ) through the project Boosting Uganda's investment in livestock development (BUILD) (Grant number BMZ001) and One Health Research, Education and Outreach Centre in Africa (OHRECA) (Grant number BMZ002). Additional time support was received from the CGIAR Initiative Sustainable Animal Productivity (SAPLING) which is supported by the contributors to the CGIAR Trust Fund. (https://www.cgiar.org/funders). There was no other additional external funding received for this study. The funders had no role in study design, data collection and analysis, decision to publish, or preparation of the manuscript.

**Competing interests:** The authors report that there are no competing interests to declare.

## Introduction

### Gender and livestock management

Livestock contribute one third of the global agricultural value output in low- and middle-income countries [1]. For the impoverished, livestock are an essential resource, particularly for women who have easier access to and control over small livestock such as sheep, goats, pigs, and poultry compared to land, cattle, and other material or monetary assets [2,3]. In Uganda, many depend on livestock for their livelihoods, especially pastoralists, agro-pastoralists, and those keeping animals in mixed crop-livestock systems [4,5]. Social and gender dynamics influence the use of livestock innovations [6], including disease control practices, with variations by livestock species kept and production system [7]. Gendered differences in livestock ownership, management, and income control contribute to women's and men's access to and use of innovations [8]. Although women may claim ownership of livestock, their ability to make decisions regarding the sale of animals is often limited [7]. Across production systems, women traditionally play a prominent role in feeding animals, fodder collection, and cleaning animal sheds, whereas men focus on vaccination and breeding [9,10]. In pastoral societies, women's participation in livestock management is mostly within the homestead while men oversee general herd management away from the home, including the marketing and treatment of sick animals [11,12]. In mixed crop-livestock systems, gender roles in livestock management vary across regions. Women collect fodder, feed and water animals, clean sheds, and milk, whereas men are usually in charge of the sale of animals. Children, especially boys, are responsible for herding [13]. In agropastoral production systems, decision-making largely depends on the value and purpose of animals and their products. For example, if the purpose of the animal is within the woman's responsibilities such as feeding the family, her decision-making is greater than for draft animals, because plowing is an activity undertaken primarily by men [2,13].

Men and women may prefer different strategies for prevention and control of livestock disease. The choices are influenced by the availability of money, time, labour, information, cost of technology, and hierarchical power relationships within the household/community [14]. Women and men tend to have different authorities and responsibilities regarding animal management [15] and these roles impact disease control practices. Research from other African countries, such as Ghana and South Africa, has shown that access to resources and decision-making power within the household or community influenced the participation of men and women in particular agriculture practises [3,16,17]. However, there is still limited knowledge on how gender dynamics interact with and influence veterinary disease control practices.

Previous studies have explored key issues including gendered division of labor, livestock ownership, decision-making, and constraints in accessing services [18]. A study in Senegal focused on gendered barriers within vaccine distribution chains [19]. In Uganda, Acosta et al. [20] identified vast inequalities in access to peste des petits ruminant (PPR) vaccines, information, and access to acaricides; identities associated with the most disadvantage included being female, elderly, or living in a

remote area. Acaricides have been reported to have adverse effects on human health if ingested or inhaled [21]. Given the high incidences of domestic violence against women in Karamoja [22], and cases of women using acaricides to poison themselves to escape domestic abuse, women have been banned from purchasing acaricides [20]. Another study in Uganda and Kenya by Mutua et al. [23] highlighted gendered barriers limiting RVF vaccine uptake, finding that married women had more decision-making authority for treatment than for preventive veterinary services [24]. This study builds on existing knowledge to determine the factors that influence disease control practices for men and women across three production systems in Uganda with the goal of identifying tailored and effective strategies for the control of PPR and RVF.

## PPR and RVF in Uganda

Diseases are a major constraint for livestock production systems [25]. PPR and RVF are viral diseases that impact livestock keepers in Uganda [26–28]. PPR is a transboundary disease that affects small ruminants (SRs) [29]. It can cause mortality from 23–100% in naïve flocks, however in endemic areas such as northeastern Uganda (Karamoja region), PPR is more likely to cause reduced milk yields, poor body condition, and secondary bacterial infections. Recent reports have described PPR extending into non-endemic districts in central and southwestern Uganda [26]. Husbandry practices such as seasonal communal grazing and watering have been reported to increase PPR transmission [26,30]. Strategies for controlling PPR include livestock vaccination, movement restrictions, isolation of sick and new animals, and safe disposal of animals that have died from the disease [29]. To guide the control and eradication of PPR in Uganda, a national strategy was created that makes references to several existing policies including the Animal Diseases Act, Cattle Grazing Act and Cattle Traders Act [31]. There are no acts with a specific focus on small ruminants; however, a national strategy was developed to guide the control and eradication of PPR in Uganda [31], which references existing policies including the Animal Diseases Act [32], the Cattle Grazing Act [33], and the Cattle Traders Act [34].

RVF is a mosquito-borne zoonosis that causes abortions and death in ruminants including cattle, sheep, and goats [35]. When people are infected, symptoms range from mild flu-like illness to severe illness including meningoencephalitis, hemorrhagic fever, and death [36]. RVF infection has also been associated with miscarriages in women [37]. Seroprevalence in domestic ruminants in the south-western districts of Uganda was estimated to be 15% in a 2016 study [38]. RVF can be controlled through vaccination of livestock, movement restrictions, personal protective equipment for people handling sick animals or slaughtering, safe handling of livestock abortions, and vector control [28]. Currently, there are no government guidelines on livestock vaccination for RVF in Uganda [27].

When PPR and RVF spread among livestock herds, losses include animal mortalities (including abortions); reduced income from the sale of live animals, milk, meat, and manure; and the loss of livestock as an investment [39]. Understanding disease prevention and control measures taken up by men and women to control the two diseases different production systems can support policy makers in designing targeted disease control interventions. The objective of this qualitative study was to identify what men and women in different production systems do to prevent or control diseases that affect them and their livestock and what factors influence the choice of disease control measures taken.

## Methodology

This qualitative study used sex-disaggregated focus group discussions (FGDs) with livestock keepers and key informant interviews (KIIs) with experts to identify factors influencing disease control practices for men and women in Uganda across three production systems (pastoral, agro-pastoral, and mixed crop livestock). The study tools and data analysis were informed by two frameworks: the gender analysis framework (GAF) and socioecological model framework (SEM). The GAF considers the roles of men, women, the elderly, and children, their access to resources, and the socioeconomic context [14]. The SEM emphasizes people's interactions with their physical and socio-cultural environment [40], considers behavior to be influenced by multiple levels (individual, interpersonal relationships, the community, and policy), and can be used to support behavior change [41,42]. SEM has been extensively applied in the field of human public health [43–47]

but rarely in animal health [20]. Adopting the SEM in the context of livestock disease management provides a different perspective and allows the analysis of the influencing factors at each level from individual behaviors to broader policy issues. Integration with the GAF helps to understand gender roles in livestock disease management, differential access to resources/services, and how structural factors shape the roles and access to the resources/services [48].

## Description of study sites

Data collection activities were conducted between December 2020 and May 2021 in six districts of Uganda: Isingiro, Nakapiripirit, Serere, Napak, Sembabule, and Butebo (Fig 1). These districts are all located in the cattle corridor which stretches from the southwestern to the northeast of Uganda. The cattle corridor hosts 90% of the country's cattle in rain-fed agro-pastoral and pastoral production systems and is vulnerable to climatic shocks [49].

To support the goal of understanding diseases control practices for both PPR and RVF, the project purposively selected Isingiro, Nakapiripirit, and Serere as PPR sites and Isingiro, Napak, Sembabule and Butebo as RVF sites. These districts were chosen because of their higher prevalence of disease based on epidemiological reports describing the risk of PPR in small ruminants [50–53] and RVF in animal and human populations [38,54–56]. Isingiro in western Uganda and Butebo in the east are largely agropastoral districts with indigenous, cross, and exotic livestock breeds. Napak and Nakapiripirit districts in northeastern Uganda and Sembabule district in the southwest are mainly pastoral with some elements of agro-pastoralism. Pastoral systems are characterized by arid conditions where grazing by ruminants is the predominant form of land use. Livestock keepers raise indigenous breeds, with herd sizes ranging from a few to 100 heads of animals [4]. Pastoral systems often require livestock keepers to make long-distance migrations in search of pasture and water during droughts. In agropastoralism, animals range over short distances and livestock production is associated with dryland or rain-fed cropping [57].

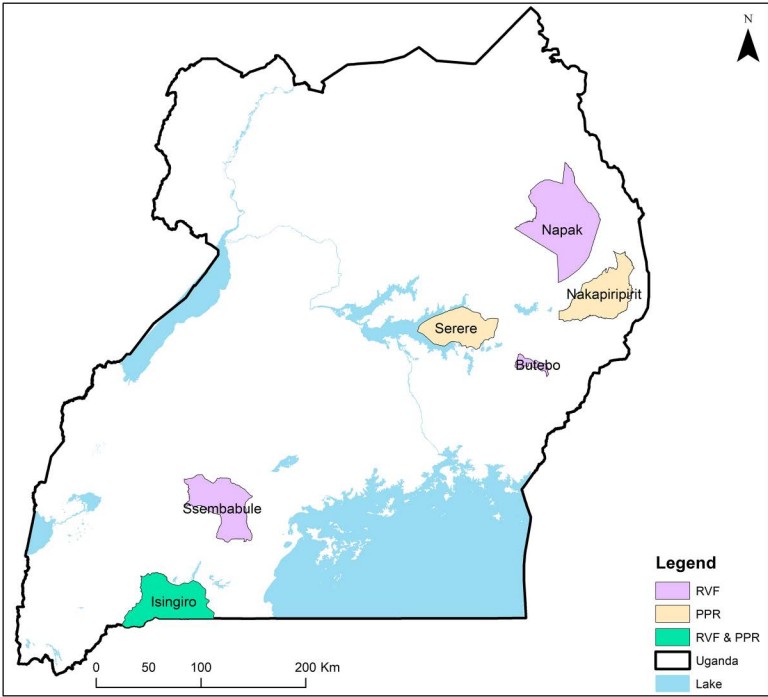

**Fig 1. Map showing the study districts within Uganda.**

In Nakapiripirit, there are frequent livestock interfaces with wildlife, which can contribute to the transmission of diseases to livestock. Butebo and Serere districts in eastern Uganda have mixed crop-livestock production systems, limited grazing land, and many livestock keepers practice zero grazing. The production system in the eastern region is characterized by both traditional and commercial dairy production [58].

**Study design, sampling, and data collection.** The guides for the focus group discussions and key informant interviews were designed to explore the factors influencing the disease control options taken up by men and women. Guided by the GAF, content in the guides included identifying the types of livestock owned or kept by men and women, gender roles in livestock management, disease prevention and control, gendered access to information and veterinary services, and factors that influence the patterns of activities, access, and control of resources [14].

The SEM also contributed to the design of the study guides through questions to address factors at five levels that may influence disease prevention strategies used by livestock keepers (the names of the levels appear in bold). **Intrapersonal** factors were addressed through questions on attitudes, beliefs, perceptions, and knowledge about PPR/RVF. **Interpersonal** factors included social influences ranging from interactions with animal health workers, fellow livestock keepers, social functions in the community, family responsibilities, and aspects of decision-making. **Community** factors explored included veterinary/vaccination information outreach and feeding practices at community level, perception of livestock disease risk and outbreaks. **Institutional** factors included vet service costs, distance to service centers, availability of animal health providers, information access, and socioeconomic inequalities between livestock keepers. **Policy** factors were explored using questions regarding the rules and regulations surrounding PPR/RVF prevention and control. The FGD and KII guides are provided in Supporting Information (S1 and S2 Files, respectively).

**Focus group discussions.** In total, 28 FGDs were conducted: 16 FGDs with women and men keeping cattle in the RVF districts, and 12 FGDs with women and men keeping small ruminants in the PPR sites. The numbers were slightly different because the RVF sites had four enrolled districts and the PPR sites had only three, with Isingiro District selected for both diseases. The sampling protocol for the FGDs was as follows: two sub-counties from each study district (Fig 1) were randomly selected for hosting research activities. In each subcounty, one parish was selected, and eligible livestock keepers were randomly selected for invitation to participate. A list of livestock keepers with cattle and SRs was provided by the district production office for all participating sub-counties/parishes and used as a sampling framework. The list included gender, marital status, and livestock species reared for each livestock keeper. In the PPR sites, the sampling strategy targeted livestock keepers with small ruminants, whereas in the RVF sites, livestock keepers with cattle were targeted. Of course, many livestock keepers in the study sites kept both small ruminants and cattle, and PPR or RVF could occur in any of the study sites. Isingiro District was targeted for both PPR and RVF because of the reported occurrence of these two diseases [26,38]. For each sub-county/parish, a women's and men's FGD was held with six to eight discussants each. Discussants in the women's FGD were drawn from male- and female-headed households in equal proportions. This was done to acknowledge the different circumstances of married women versus women heads of household that likely influence their constraints and decision-making ability regarding disease management.

Prior to conducting the research activities, a team of four people fluent in the respective local languages (two elders and two animal health practitioners from each study district) translated the study guide. The translated guide was further refined by local facilitators from each district during a two-day training workshop. The enumerators were taken through the questions and associated probes to ensure that they understood the purpose and intent of each question. The FGD guide was then locally pretested with two sex-disaggregated groups of livestock keepers.

Before the start of each FGD, each discussant was individually asked for demographic information including gender, age, marital status, household headship, and education level. Each FGD explored questions on grazing/watering resources for SR/cattle within their community, access to resources, constraints, awareness of SR/cattle diseases (how these diseases were recognized, possible causes, and perceived risks of exposure to animals/humans), disease management/control practices and access to animal health services/vaccinations. Additional questions on zoonoses, risk, and

mitigating practices, such as vector control, were asked in the RVF sites. The FGD sessions were moderated by a local facilitator and documented by a notetaker who understood the local language of the study site. The local facilitators and note takers were supervised by one of the Ugandan co-authors. Using the FGD guide, the trained local facilitators led the FGDs in the local language of each district while the notetakers took notes verbatim. Men facilitated men's groups while women facilitated women's groups. The sessions were conducted in locations such as sub-county offices, community halls, and school premises and participants were provided with lunch and refreshments. Since the FGD guide covered many topics, the sessions lasted for half a day. To minimize on research participant fatigue, participant engagement was maintained through participatory exercises such as simple ranking, resource mapping, and proportional piling [59,60]. Simple ranking and proportional piling were used to determine the relative importance of different diseases to livestock keepers. The results from resource mapping were used to generate information on resource use (grazing, feeding, marketing, veterinary services) and the availability of these resources to both men and women within these communities. Data were used to build on explanations of the factors influencing disease prevention. Information outside the scope of this article was used for other project outputs [61–65].

**Key informant interviews.** The key informant interview guide (S2 File) was designed to triangulate the findings from livestock keepers on the same topics. Similar questions were asked at all sites. Four of the co-authors (Ugandans) facilitated the KIIs in English. Some of the KII were conducted over the telephone due to COVID restrictions in some of the study regions at the time.

Key informants were chosen purposively from the study sites based on their professional experience working with livestock keepers. A total of 32 KIIs were conducted for PPR (12 men and 2 women) and for RVF (14 men and 4 women). Key informants included district production and marketing officers, veterinary officers (VOs), district veterinary officers (DVOs), animal husbandry officers (AHOs), community-based animal health workers (CAHWs), veterinary drug stockists, local council, and livestock keeper group leaders. VOs, DVOs and AHOs can also be referred to as government animal health workers (AHWs) and are usually supported by private (non-governmental) community-based animal health work-ers. The KII guide included questions for multiple types of informants, for example, CAHW, AHO, or policy maker. The KIIs were not audio-recorded, however detailed notes were taken by the research team.

**Consent and eligibility.** All participants were at least 18 years old. For KIIs conducted either over the phone or virtually, discussants provided verbal consent. Before the start of each FGD, written consent was obtained from each discussant using a consent form interpreted in the local language for each district: Karamojong for Napak and Nakapiripirit; Runyankole for Isingiro; Luganda for Sembabule; Ateso for Serere; and Luganda, Ateso, or Lugweere for Butebo.

**Data analysis.** The notes from the research activities were translated into English by local facilitators. Personal identifiers were removed from the transcribed files to ensure anonymity of responses. The transcripts were then reviewed and verified for completeness by the research team, including the local facilitators before being imported into NVivo 12. The data analysis incorporated both deductive and inductive approaches. The coding framework of key themes was developed using a deductive approach guided by the research questions and topics in the GAF, such as gender roles, whereas other themes emerged after reviewing the qualitative data and were incorporated into the coding framework in subsequent iterations (inductive approach) [66]. The GAF was applied to the first iteration of coding while the SEM was applied to the second iteration. The coding framework is available in Supplemental Information S3 File.

The initial codes included gender roles in livestock production activities, access to resources and services for dis-ease control and prevention, and ownership. The preventive and control measures were sub-coded between men's and women's measures (treatment, vector control, culling, movement control and vaccination). All coded data for the PPR and RVF data sets were read and re-read to identify connections, similarities, or differences between the codes. Based on emerging patterns and relationships, codes were organized into overarching themes disaggregated by gender and

production system. Risky behaviors predisposing people to zoonoses (specific to RVF) were an example of an emergent theme added to the coding framework later. The PPR and RVF data sets largely used the same coding themes with a few exceptions.

In the second iteration of coding, preventive and control measures were categorized into within farm, between farm and community level, depending on where the activity took place. The factors that influenced choices of preventive and control methods were also coded. This analysis focused on the underlying factors that determined the gender roles, and access to and control over resources/services. These included source of services and information, availability, accessibility, and affordability of services, perceptions/views/concerns towards a method, knowledge and awareness, socio-cultural factors, decision-making aspects, and trust/experience with a method. The aim of the analysis was to identify and understand participants' perceptions and constraints of the preventive and control measures. The selected themes were then categorized and analyzed using the SEM described by Mcleroy [41], composed of intrapersonal, interpersonal, institutional, community and policy levels. For the selected themes, the number of mentions by each FGD, disaggregated by gender, was counted and summed across production systems. Notations in the results section about how many FGDs mentioned a theme or topic were used to indicate that it was mentioned by at least one discussant in the focus group. Discussant quotes were also provided to illustrate these themes. Data from the KIIs were used to validate the accuracy of the information generated from the FGD and to build on the explanations of the FGD results. Quantitative demographic data were analyzed using the Statistical Package for Social Sciences (SPSS software program).

## Ethics statement

This study was approved by the Institutional Review Board School of Biosecurity Biotechnical and Laboratory Science (SBLS) at the College of Veterinary Medicine, Animal Resources and Biosecurity (COVAB), Makerere University (SBLS.KR.2020), the Uganda National Council for Science and Technology and the ILRI Institutional Research Ethics Commission.

## Results

### Demographic characteristics of FGD discussants

The demographic characteristics of the discussants are presented in Table 1. The results showed that more women than men reported no formal education in both the PPR and RVF sites (39% versus 16%), with greater differences in the PPR sites.

Table 1. Demographic characteristics of FGD discussants.

|  | PPR sites | | | RVF sites | | |
|---|---|---|---|---|---|---|
|  | Men | Women | Overall | Men | Women | Overall |
| # of participants | 58 | 55 | 113 | 69 | 61 | 130 |
| Mean age (years) | 45 | 39 | 42 | 47 | 46 | 46 |
| **Education level** | | | | | | |
| No formal (%) | 8 (14) * | 22 (40) | 30 (27) | 12 (17) | 17 (28) | 29 (22) |
| Primary (%) | 28 (48) | 22 (40) | 50 (44) | 25 (36) | 25 (41) | 50 (38) |
| Secondary (%) | 19 (33) | 9 (16) | 28 (25) | 24 (35) | 13 (21) | 37 (28) |
| Vocational (%) | 1 (2) | 2 (4) | 3 (3) | 7 (10) | 5 (8) | 12 (9) |
| University (%) | 2 (3) | 0 (0) | 2 (2) | 1 (1) | 1 (2) | 2 (2) |

*Figures in parenthesis are percentages, #=Number of participants.*

## Thematic analysis

We present the results of the thematic analysis in two sections: The first section uses GAF, and the second section uses SEM. The SEM section considers prevention and control measures for ruminant disease (within farm, between farm, and at the community level) and factors influencing disease prevention and control (intrapersonal, interpersonal, institutional, community-level, and policy-level). The results show that certain gender roles in livestock management are consistent across production systems. Both women and men recognized that traditionally, cattle were primarily owned by men who served as heads of households, with exceptions for women who were heads of their households. Men frowned on the idea of women owning cattle. Most women could not afford the hefty expenses of purchasing, feeding, and veterinary care. Women and children could own a few goats or sheep, but not as many as men. Women's and girls' tasks were limited to caring for animals within the homestead, such as cleaning animal sheds, caring for calves/kids/lambs, pregnant animals, and tethering animals for grazing. Men and boys grazed animals farther from home in search of pasture, feed, and water from distant areas during droughts.

## Prevention and control measures for ruminant diseases

The prevention and control measures mentioned by men and women across the three production systems are presented in Table 2. In general, more disease management measures were implemented for cattle than for SRs, and vaccination was rarely mentioned for SRs. Women across production systems reported more measures than men, and more measures for SRs (Table 2). Most prevention and control measures were implemented within the farm, with vaccination being

**Table 2. Prevention and control measures across three production systems as reported by men and women.**

| | Small ruminants | | | | | | Cattle | | | | | |
|---|---|---|---|---|---|---|---|---|---|---|---|---|
| | Pastoral | | Agropastoral | | Mixed | | Pastoral | | Agropastoral | | Mixed | |
| | M | W | M | W | M | W | M | W | M | W | M | W |
| **Within farm** | | | | | | | | | | | | |
| Identify sick animals | | ✓ | ✓✓ | ✓✓ | ✓ | ✓✓✓ | ✓✓✓ | ✓✓ | ✓✓ | ✓✓✓✓ | ✓✓ | ✓✓✓ |
| Isolate sick animals | ✓✓ | ✓✓ | ✓ | ✓ | ✓✓ | ✓✓ | ✓ | ✓✓✓ | ✓ | ✓✓ | | ✓✓✓ |
| Consult AHWs | | ✓ | ✓ | ✓✓ | ✓✓ | ✓ | ✓ | ✓✓✓ | ✓✓✓✓ | ✓✓✓✓ | ✓✓ | ✓ |
| Traditional medicine | ✓✓ | ✓ | | ✓✓ | ✓ | ✓ | ✓✓ | ✓ | ✓✓ | ✓✓ | ✓ | ✓✓ |
| Apply acaracides | | ✓ | ✓ | ✓✓ | ✓✓ | ✓✓ | ✓✓ | ✓✓ | ✓✓✓✓ | ✓✓✓✓ | ✓ | ✓✓ |
| Deworm | | ✓ | | ✓✓ | | ✓✓ | | | | ✓✓ | | ✓✓ |
| Cull sick animals | | | | ✓ | | | | | ✓ | ✓ | | |
| Clean enclosures | | | | | | | | ✓ | | | | |
| Slaughter or sell | | | | | | | | | ✓ | ✓ | | |
| Veterinary drugs | | | | ✓ | | ✓ | ✓ | ✓✓✓ | ✓✓ | ✓✓✓ | ✓✓✓✓ | ✓✓✓✓ |
| **Between farms** | | | | | | | | | | | | |
| Fence farm | | | | | | | | | ✓ | | | ✓ |
| Relocate kraals | | | | | | | ✓ | ✓✓ | | | | |
| Controlled grazing and watering | ✓ | | | ✓ | | ✓✓ | ✓✓ | ✓ | ✓✓ | ✓ | ✓✓✓ | ✓✓✓ | ✓✓✓ |
| **Community** | | | | | | | | | | | | |
| Vaccination | ✓ | | | | ✓ | | | ✓✓ | ✓✓✓ | ✓ | | ✓✓ |

*One tick (✓) = The number of FGDs in which at least one participant mentioned the respective measure, M = Men, W = Women, AHWs = Animal health workers.

Mixed = Mixed-crop livestock production system.

the only community-level measure mentioned. In agropastoral areas, livestock keepers reported consulting with AHWs and CAHWs before administering purchased veterinary drugs themselves in an attempt to control PPR.

**Within farm prevention and control.** Prevention and control interventions within farms included prophylactic measures, vector control, culling, and treatment. The involvement of men and women in the identification of sick animals, isolation of sick animals and cleaning of animal enclosures varied across production systems and livestock type reared. In mixed crop-livestock production systems, more women than men were involved in identifying and isolating sick animals. A DVO from Serere District remarked that very few men took on these roles, even in households with large numbers of SRs. In the pastoral households of Napak, women checked for signs of sickness while livestock, especially small ruminants, were in the *kraal* (livestock enclosure), then informed their husbands, sons, or male employees who herded animals. Across all systems, women helped to isolate sick animals from healthy ones. In both agropastoral and pastoral areas, women were traditionally viewed as caretakers of the home and were therefore responsible for managing livestock within the homestead. This included pregnant, sick and young animals, making it likely for women to observe abortions. In mixed crop-livestock production systems, both men and women isolated sick animals, especially cattle. In the agropastoral areas of Isingiro District, some farms were managed and controlled by women when their husbands were away. In pastoral areas, women cleaned the *kraals* and reported disease cases to CAHWs, especially for SRs. Girls helped their mothers clean animal enclosures.

Some livestock keepers routinely sprayed all their animals with acaracides, both SRs and cattle, whenever they noticed any signs of sickness or tick accumulation. In agropastoral households, spraying with a backpack sprayer was sometimes a joint family activity. In mixed production households, women considered men more knowledgeable about acaricide use; therefore, men took on the responsibility of purchasing and mixing acaricides. In pastoral areas, livestock keepers typically sought the services of CAHWs and utilized community crushes. During the dry season, finding water for mixing acaracides was sometimes a constraint. In mixed production systems, women mostly reported seeking the services of AHWs to spray their animals while some men sprayed by themselves. Men in the FGDs in Napak and Nakapiripirit described boys and girls removing ticks by hand. Women were reported to have more contact with both male and female CAHWs than men, seeking to help spray their SRs whenever they noticed tick infestations. Additionally, women bathed the kids and lambs with powdered washing soap or dusted them with purchased pyrethroids to remove ticks due to a lack of backpack sprayers. These options were less expensive than purchasing acaracides

Only three FGDs (one with men and two with women), all in agropastoral production systems, reported culling as a disease control measure. If an animal was too sick, they would either sell or slaughter it and eat the meat. In pastoral systems, if the condition of the SR worsened, men slaughtered the sick animal for home consumption or sold the animal to avoid disease spread within their farm because burying or throwing it away was considered a waste. An AHW from Nakapiripirit noted that when SRs get sick, livestock keepers let them die or slaughtered and ate the meat.

*In my home, I slaughter the sick animal or sell it before the diseases get more serious.* (Women's FGD, Nakapiripirit District, pastoral system)

Across production systems, boys identified sick animals while grazing, separated sick animals from healthy ones, then reported to their parents. In cases of sudden death, men in pastoral and agropastoral systems would open the carcass and check the internal organs to try to identify the cause of death. If any internal organs, such as lungs or liver, were affected they were removed and buried. The remaining meat from the examined animal was distributed among the neighbors and the hide was cured for mats or bedding. Nothing would be wasted, in part because animals were rarely slaughtered for home consumption. An AHW reported that people believed that meat should be eaten and that burning or burying meat was a sin. Aborted fetuses were roasted and eaten or fed to dogs.

*I would first skin the animal and try to identify the possible cause of the death. Then, the meat is eaten by the family.* (Men's FGD, Nakapiripirit District, pastoral system)

*My husband does not buy meat for us and does not want to slaughter for home consumption, yet it's my children and me who graze them, so it will be an advantage to have meat when the animal dies.* (Women's FGD, Nakapiripirit District, pastoral system)

In all production systems, local herbs were commonly used by both men and women as the first line of treatment before other options were sought. In both pastoral and mixed crop-livestock systems, these were considered effective, cheaper, and easier to access than veterinary products. Herbs were selected based on disease type using local knowledge and previous experience. For example, in the pastoral areas of Napak, women used *Combretum* spp. (willows) to treat PPR. Other local herbs included *Aloe vera* and *Azadirachta indica* (neem tree leaves). Local herbs were reported to be the most effective in the early stages of respiratory infections in pastoral areas. In mixed and pastoral systems, men were responsible for sourcing herbal medicine and treating animals.

*We source local herbs from the bush and mix [them] for the goats to drink in case of PPR. These herbs are more effective for younger animals (goats) than for older ones. This is also cheaper for us because it's easy to get in the villages.* (Men's FGD, Nakapiripirit District, pastoral system)

In the mixed crop-livestock systems, mostly in Butebo District, livestock keepers relied on local herbalists to treat sick animals, appreciating that they were accessible at any time of the day. In pastoral systems, veterinary personnel were only called upon when sick animals failed to recover. There were few AHWs and their services were perceived as expensive. The use of veterinary drugs was either through treatment by the livestock farmer or by sourcing the services of an AHW. Consultations were more common for cattle than SRs because of cattle's higher values and longer lifespans. Women often consulted AHWs on which drugs to use and for which diseases. In agropastoral systems, some men considered themselves to be experienced in the treatment of animals due to interactions with AHWs. Others consulted with experienced peers within the community, purchased drugs, and administered to their livestock themselves. Veterinary drugs were purchased from veterinary drug shops, traveling vendors, or markets depending on the locations. In pastoral and mixed crop-livestock systems (Isingiro and Serere districts, respectively), women mostly bought dewormers and men mostly purchased drugs for cattle. Men were responsible for decisions regarding the treatment of both cattle and SRs. The DVO in Serere (mixed crop livestock system) reported that it was rare for men to call AHWs about sick goats and that women struggled to restrain the goats together with the children when receiving animal health services.

*I do the treatment by myself since government AHWs are scarce and I can't get the money to pay the bills.* (Women's FGD, Nakapiripirit District, pastoral system)

**Between farm prevention and control.** Men in the mixed and agropastoral systems believed that direct contact led to spread of tick-borne diseases to their livestock, and as a result, took action to restrict animal movement to avoid contact with livestock from neighboring farms. This included digging trenches to deter visiting livestock or fencing their farms. Fencing was not always a feasible option because of the costs involved in buying poles and the labor required to maintain the fences. Fencing individually owned farms at the household level in agropastoral and relocating animal *kraals* in pastoral areas was done more for cattle than for SRs. Women avoided mixing their livestock with neighboring livestock by tethering them while grazing, feeding them from home, and providing drinking water in separate water troughs. In the pastoral system, *kraals* were mainly communally owned, and in case of disease outbreaks, animals were moved from one location to another within the community under the stewardship of the *kraal* leader. In the case

of individually owned *kraals*, during disease outbreaks, healthy animals were separated from sick ones, moved to a different location on the same land, and continuously monitored. In pastoral systems in Napak District, men instructed the boys not to take animals to grazing land where other sick animals were grazing and to check the watering points before taking animals.

*We either tie the goats and the sheep which are sick to avoid mixing them with the healthy ones or separate them in an isolated kraal until they are vaccinated or treated, then we release them for grazing.* (Men's FGD, Nakapiripirit District, pastoral system)

**Community prevention and control.** Livestock vaccination was the only community level measure mentioned in the FGDs. Livestock keepers reported that vaccination was initiated by the livestock production authorities. The CAHW alerted community members about upcoming vaccination through phone calls to household heads, informing them about the vaccination date and the species to be vaccinated. Other communication channels used included community places of worship, posters pinned up in public areas, and household-to-household mobilization, especially in mixed production systems. As heads of households, men were reported to be the key decision-makers regarding vaccination of the animals. Across production systems (eight men's FGDs, five women's FGDs), cattle were more often vaccinated than SRs. FMD was the main disease vaccinated against. Consequently, some discussants were familiar enough with FMD to refer to it directly using the abbreviation, as shown in the quotes below.

*If it is not FMD, do not expect any mass vaccination.* (Men's FGD, Serere District, mixed crop-livestock system)

*We have only gotten FMD vaccines for cattle since animal health workers were not considering small ruminants in this village.* (Women's FGD, Nakapiripirit District, pastoral system)

Vaccination against PPR was reported only in the pastoral production systems. None of the study sites mentioned having ever vaccinated against RVF and were not aware of RVF vaccines. Animal health workers across production systems reported that none of the veterinary drug shops stocked vaccines because they lacked freezers and equipment for appropriate transportation, storage, and distribution.

## Factors influencing disease prevention and control options taken by men and women

This section summarizes the factors influencing the choice of disease prevention and control measures, disaggregated by gender and production system. Following the SEM model, we consider factors at multiple levels: intrapersonal, interpersonal, institutional, community, and policy. While livestock keepers embraced disease control options, such as use of drugs, spraying acaracides, and use of traditional herbs, many had concerns and misconceptions about vaccination as a preventive measure, reflected in all levels of the SEM (Fig 2).

**Intrapersonal.** Intrapersonal factors included concerns about vaccine side effects, disease resistance and suspending consumption of meat or milk after treatment. Vaccine side effects were the most frequently mentioned factor. Men, particularly in pastoral communities, were more concerned about vaccine side effects than women. Their concerns included reduced fertility of animals, neck swelling, restlessness, abortion, lameness, death, and tail drop in cattle. In agropastoral and mixed crop livestock systems, women's limited knowledge of livestock diseases, for which vaccination mobilization is conducted, prevented them from adopting vaccination as illustrated:

*Unless we are sensitized about the diseases, we will assume our animals are not infected with the diseases. Training or sensitization through videos showing disease signs could enable us to visualize the diseases.* (Women's FGD, Sembabule District, agropastoral system)

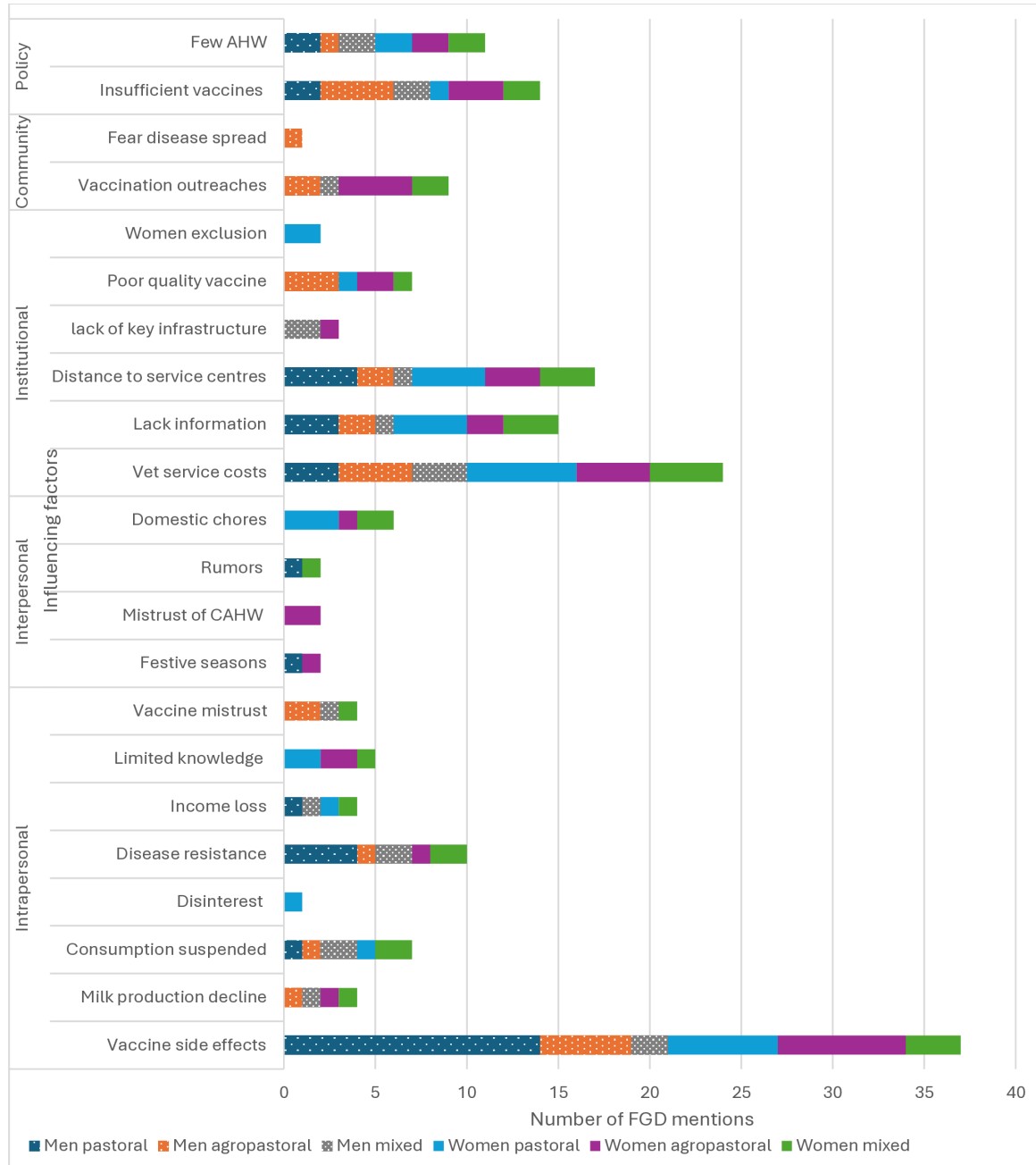

**Fig 2. Factors influencing disease control option uptake by gender and production system.**

In the pastoral system of Napak and mixed crop livestock system of Butebo, some men believed that vaccination triggered other animal diseases and increased their susceptibility to other livestock diseases. This was attributed to overdoses, expired vaccines, or incompetence of AHWs. In agropastoral areas of Isingiro District, some livestock keepers preferred farm-to-farm vaccination to avoid mixing their animals with others from different locations, fearing risk of animal theft, fights and injury amongst animals especially bulls, and disease transmission. Some women in the mixed crop

livestock of Butebo District and pastoral systems of Napak District believed that vaccinating their animals would make them more resistant to diseases, thus reducing mortality rates and increasing milk yields. Women, particularly in Isingiro District, voluntarily contacted an AHW to vaccinate their animals after confirmed outbreaks in the village because they were afraid of losing their animals. Men from the same district regarded vaccination of cattle as a way of earning respect from fellow livestock keepers and a way to fetch higher prices at the livestock market. A veterinary drug stockist from Nakapiripirit reported that many livestock keepers do not differentiate between drugs for treatment and vaccination for prevention. The misconception that vaccination requires a withdrawal period before meat or milk can be consumed hindered some women in the mixed crop-livestock system from embracing vaccination. Some men in the same system feared the meat from vaccinated animals would have an unpleasant taste.

*My income is in milk, if I take (animals for vaccination), I may worry about the source of income for the period I will not be milking.* (Women's FGD, Butebo District, mixed crop-livestock system)

**Interpersonal.** The interpersonal factors included rumors, mistrust of AHWs, household roles and expectations, and socio-cultural functions (such as marriage, initiations, and Christmas). Based on these findings, disease control options for women were more influenced by social expectations concerning domestic responsibilities. Women found it difficult to participate in vaccination programs due to heavy domestic workloads such as cooking and taking care of children, which limited their movement beyond the domestic sphere, a concern not mentioned by men.

*In our community, women are the ones to sweep where the goats sleep. If you do not clean where your ruminants sleep, no one will come and clean it for you. This helps reduce the odor from the accumulated dung and prevents disease from affecting the animals. Men do not have the time for such things.* (Women's FGD, Serere District, mixed crop livestock system)

In pastoral areas, rumors reported from neighboring sub-counties about the side effects resulting from vaccination of SRs made some men hesitant to participate in vaccination campaigns, whereas some women presented their SR for vaccination for fear of being accused of spreading diseases in the village.

*I would take [animals] in case of [an] outbreak so I would not be accused in the community because my animals were not vaccinated.* (Women's FGD, Nakapiripirit District, pastoral system)

In the pastoral and agropastoral areas of Napak, Nakapiripirit, and Isingiro districts, it was difficult for livestock keepers to participate in vaccination during festive seasons. Some livestock keepers did not trust the AHW services as illustrated below:

*I would not take my animals for vaccination. This is because most of these CAHWs do not know the right vaccines for some diseases. So, they may kill my animals if they vaccinate with the wrong vaccine.* (Men's FGD, Nakapiripirit District, pastoral system)

*What I fear most is the veterinary doctor. One time they treated my goats. One became sick, two died, and a pregnant one became lame. Since that time, I hate vets I don't know and [I am] not sure of.* (Women's FGD, Sembabule District, agropastoral system)

In the agropastoral areas of Isingiro and Sembabule districts, livestock keepers preferred seeking services from trusted private or government AHWs because of the availability of treatments on credit and the comprehensive stock of necessary drugs.

**Institutional.** Institutional factors included veterinary services costs, distance to animal health centers, few AHWs, lack of key infrastructure such as animal crushes, poor quality animal health products, women's exclusion from extension services such as trainings, and limited information sharing. Across production systems, the small number of AHWs was reported as a challenge, as described below.

*In Kanginima sub county, we have one animal health worker, but he has no drug shop, he moves with drugs. Thus, local herbalists and community animal health workers are more accessible.* (Men's FGD, Butebo District, mixed crop livestock system)

*It is not easy to get a vet officer because they do not stay at the sub-county level, and they are lone workers. You can call the animal health worker in the morning, and he/she reaches you in the evening. This worsens the health of the sick animal. That is why we treat our animals by ourselves.* (Men's FGD, Isingiro District, agropastoral system)

*It's rare to access government vets since they only come for vaccination, and this is the only time my goats get treated.* (Women's FGD, Nakapiripirit District, pastoral system)

Despite community mobilization, vaccination was always associated with a cost in terms of transporting animals, compensating the AHWs for their transport to the communities, or restraining animals at the cattle crush. For women whose husbands were absent, an additional cost involved hiring help to move animals to the vaccination point. The transport facilitation charges for the AHW, which varied between UGX 500 to UGX 5,000 ($0.14 – $1.40 USD) per animal depending on the distance to the vaccination point [67], were a major hindrance to both men and women. Across production systems, announcements about vaccination of animals were reportedly at short notice with limited community outreach. Livestock keepers often failed to raise the money on short notice to pay for the charges requested, making them resent the entire program. In some locations, AHWs prioritized wealthier livestock keepers over smallholders.

*They usually say it's free, but if I have 100 cattle, that is UGX 200,000 [$ 52.64 USD] which is very expensive. That is why some livestock keepers do not bring their animals. The UGX 2,000 [$0.53 USD] is per cow. They should tell us when there's still time so that we can look for money.* (Men's FGD, Sembabule District, agropastoral system)

*When it's time for vaccination, AHWs prioritize the rich and those who have large herds. They always say there are no doses for people with small herds. Those with small herds end up not benefiting.* (Men's FGD, Sembabule District, agropastoral system)

Vaccination points were chosen in collaboration with local leaders and DVOs. Local leaders then mobilized livestock keepers to congregate their herds at the designated locations. In pastoral areas, the presence of a communal crush dictated the choice of the vaccination site. Both men and women in this production system found it difficult to trek herds to these communal crushes, particularly in remote areas. In this system, under the instruction of their fathers, young boys trekked the animals to the communal crushes. Where there were no crushes, livestock keepers had to restrain their animals, which was particularly difficult for women. During the vaccination event, the AHWs were overwhelmed and found it challenging to vaccinate all animals that were brought. In some agropastoral areas, farms were reported to be extremely far, waterlogged, and impassable during rainy seasons. This made it difficult for AHWs to reach livestock keepers.

*It is hard to access veterinary health services because the drug shops are very far from here requiring high transport costs. In addition, the drugs are expensive for us to acquire. This makes it very hard for women to access animal health services. What we do, we go to the veterinary doctor, explain the condition of the animal, and then he gives us the drugs. We come home and inject it because in most cases we cannot afford money to bring the veterinary doctor to the farm.* (Women's FGD, Isingiro District, agropastoral systems)

**Community-level.**  Community-level factors influencing disease prevention and control were related to accessibility of animal health related information and perceptions about livestock disease outbreaks. The communication channels for animal health-related information described by livestock keepers were primarily informal, including local leader announcements, places of worship, fellow livestock keepers, politicians, bars, *kraal* leaders, weekly markets, and veterinary drug shops where men converged. *Kraal* leaders played a crucial role in pastoral systems by passing on relevant information during morning hours when the herders count their livestock before taking them for grazing. Being far from such places, women normally missed such information because they were occupied with domestic chores at home, such as cooking, caring for children, or vegetable farming.

*It's men who are always in the kraal and sensitization always take place there*. (Women's FGD, Napak District, pastoral system)

Formal communication channels included AHWs, radio, and television. The low frequency of radios in remote pastoral areas made listening to the radio difficult, and as a result, critical information on disease outbreaks broadcasted over the local radios was likely to be missed. In both pastoral and agropastoral systems, women reported being left out during vaccination campaigns because organizers believed that keeping animals was only for men. In pastoral areas, women without cattle were less interested in knowing about livestock vaccination.

*Some women do not have cattle and thus have no interest in knowing anything about vaccination. Women are considered a minority group. Even if a woman buys cattle, a man takes it like it is his own, and it becomes hard for a woman to know about this information. She has no right over her own cattle*. (Women's FGD, Napak District, pastoral system)

*When the vaccination program is on, the coordinators tell people to gather all their animals in one place. These places are far. Sometimes the coordinators do not tell us women; they think that keeping animals is only for men*. (Women's FGD, Sembabule District, agropastoral system)

Across production systems, women mostly accessed information through announcements made at places of worship, on market days, and through fellow livestock keepers. During market days, CAHWs informed livestock keepers about any animal health care related services, including livestock vaccination. Information disseminated through places of worship was limited to attendees. In the mixed crop-livestock systems of Butebo District, information about vaccination was reported through newspapers, mostly targeting the literate livestock keepers, but leaving out those with low literacy levels (Table 1). Women mostly relied on CAHWs and weekly markets to access information and drugs. A male animal husbandry officer from Nakapiripirit reported that it was easier to relay veterinary/vaccination-related information to men than to women because men usually meet in town centres. Relaying information to women required following them to their homes.

**Policy-level.**  Key policy factors included insufficient vaccines and few AHWs. National level livestock disease control measures were related to scheduling mandatory livestock vaccinations for diseases such as FMD and PPR, but no such initiatives existed for RVF. The vaccination programs were reported to be infrequent, limited to a few locations, and unable to vaccinate livestock populations within the communities.

*Sometimes the AHWs conduct vaccination in a few areas and not in others. When you go to ask, they will tell you that the vaccine is finished*. (Women's FGD, Isingiro District, agropastoral system)

According to AHWs from Isingiro and Sembabule, internal restrictions have been reported on the marketing of livestock and animal products during confirmed disease outbreaks. Across production systems, FMD was mentioned as a major livestock disease, in part because outbreaks triggered movement restrictions across districts.

*Due to the cross-border movements of cattle, there have been lots of FMD outbreaks. Therefore, the Ministry of Agriculture, Animal Industry and Fisheries in conjunction with the local government decided to close all livestock markets. So, livestock trading is done at farm gate.* (VO, Isingiro District, agropastoral system)

*Right now, we're in quarantine due to an FMD outbreak. There is no movement of animals or animal products.* (Men's FGD, Sembabule District, agropastoral system)

Quarantines were instituted by the government to prevent reported livestock diseases from spreading to other areas. The closure of livestock markets affected both men and women. It led to loss of income from livestock sales, which in turn limited their ability to pay school fees, medical bills, and other necessities.

## Discussion

This study investigated what men and women did to prevent or control diseases that affected them and their livestock as well as the factors that influenced the choice of disease control measures. The findings indicate that livestock management roles strategically positioned men, women, girls, and boys to observe certain clinical manifestations of diseases. For example, boys and young men grazing animals may notice certain clinical signs, while women may notice other signs as primary caretakers for SRs, pregnant, and sick animals. Since men were more involved in culling or slaughtering of animals, they were more likely to identify clinical signs that manifested in internal organs. The roles that men, women and children play in livestock management have been widely investigated [9–13,65,68]. Given the limited knowledge of PPR [65] and RVF [69], especially among women, early disease detection, prevention and control may also be hindered. Other studies have shown how livestock keepers' lack of understanding of PPR may impact PPR prevention and control [70–72].

Of the three levels of disease control measures considered (within farm, between farm, and community-level), most of the measures described by livestock keepers were within-farm, suggesting that livestock keepers implemented control and preventive practices within their capacity. In line with other research studies [73], adoption of biosecurity practices which can reduce the risk of disease spread [74] was amongst the more accessible disease control options for women since they were mostly performed within the homestead as part of their traditional roles related to hygiene and required minimal resources. The findings on livestock keepers purchasing veterinary drugs and acaracides to administer to their livestock themselves suggests an ease of access and corroborate with other studies done in Uganda on the supply chain of antimicrobial drugs in livestock systems [75]. Unfortunately, for viral diseases such as RVF or PPR, treatment with antibiotics is not effective, and treating with the wrong drugs is costly to livestock keepers and can increase the risk of antimicrobial resistance. Tick control, while effective for preventing tick-borne diseases [76], is ineffective for control of PPR or RVF.

Although livestock vaccination is an effective way to prevent RVF [28] and PPR [29,77], it had limiting factors at multiple levels. At the individual level, fear of side effects and misconceptions about vaccines and their use was more commonly observed among men, especially in pastoral areas. As men are the primary decision-makers [78,79], their reluctance could lead to lower vaccination rates for both SRs and cattle. Lower vaccination rates could result in a higher susceptibility to PPR and RVF, leading to more frequent and severe outbreaks. Women, who have less influence over vaccination decisions, might face greater challenges in protecting their livestock, particularly SRs, exacerbating existing gender inequalities. Without prevention through vaccination and proper treatment, livestock diseases are likely to spread more easily within and between farms, thereby affecting larger livestock populations.

At the interpersonal level, misinformation from neighboring communities or fellow livestock keepers who had experienced vaccine side effects highlighted the important role of social networks. Similarly, in pastoral areas where livestock are a main source of livelihood [80], the belief that vaccines require a withdrawal period is a limiting factor for their uptake. A study in Ethiopia highlights knowledge gaps and misconceptions among livestock farmers regarding livestock vaccine

use [81]. Although livestock vaccine use differs by livestock species [82], the low adoption rate of livestock vaccines has been attributed to vaccine cost, poor access, lack of information, poor vaccine distribution, and sociocultural factors [20,78,81,83]. However, PPR vaccines have not been associated with any negative side effects [84]. Gender studies on vaccine uptake in Kenya and Uganda have demonstrated that women were particularly disadvantaged during the vaccination processes especially when it came to moving animals to vaccination points and restraining them [23]. The interpersonal level factors also reflect hindrances to women's engagement in disease control options outside the domestic sphere. Interestingly, the role of peer influence, that is acceptance and approval from fellow livestock keepers, shaped disease control options for women in pastoral systems and men in agropastoral systems. Given the patriarchal system in the production systems [85], ownership of livestock and decision making on vaccination was predominantly controlled by men. Other studies show that women can gain access to critical information and navigate the challenges of social norms through community groups [86–88]. This study demonstrates that by seeking support from fellow livestock keepers, women could gain more knowledge/information/increased awareness and confidence in managing livestock health. This is likely to foster cooperation among livestock keepers, leading to more effective disease control and prevention strategies. There might be resistance from some men who are accustomed to making these decisions, thus requiring careful navigation and support from the community.

At the institutional level, the low trust in the competency of AHWs, and the high cost of veterinary services contributed to reliance on within-farm strategies to treat and prevent livestock diseases. The same sentiments have been observed in previous studies, where the high cost of veterinary services and inadequate number of veterinary officers are drivers for livestock farmers to treat their animals without the guidance of a veterinary officer [89]. Research on veterinary service delivery in Uganda has also found staff in the veterinary departments to be inequitably distributed across the country [90]. Similarly, the absence of key infrastructure such as crushes was more of a limiting factor for women when treating or vaccinating their animals [23]. Without proper infrastructure, women may struggle to restrain animals themselves, increasing the likelihood of people or animals being injured. Women's restricted access to monetary resources might prevent them from meeting transport and vaccination related administration costs, forcing them to spend more time and effort transporting their animals to other communities where infrastructure such as crushes are available. This could be physically demanding and time-consuming. Other studies on farmer-desired attributes for a PPR vaccination campaign have demonstrated that the distance to vaccination sites is a limiting factor for women [63].

Although RVF vaccines are available in other East African countries such as Kenya [23], no guidelines on the use of RVF vaccines have been provided in Uganda [27]. Nevertheless, understanding the deterring factors for other livestock vaccines could guide future introduction of RVF vaccine. The inequitable vaccine distribution, which favored those with large herds of livestock and personal relationships with AHWs, disadvantaged individuals with small herds of livestock, especially women. This finding aligns with previous studies where the size of the herd was a determinant to access to vaccines [20] and off farm income sources a determinant to adoption of vaccination and spraying of cattle [91]. A similar observation has been noted in a previous study in Uganda showing how women as small holder farmers encountered gender specific challenges in obtaining essential extension services due to prevailing gender biases [81]. The findings also reflect the gendered access to veterinary information among livestock keepers. A study on the delivery of veterinary services in Uganda shows untimely flow of information, especially in pastoral areas, which can prevent access to veterinary services such as vaccination [90,92]. Another study in Ethiopia and Kenya, highlighted normative constraints that limit women from interacting with animal health service providers and accessing animal health information [20,93]. The insights from this study facilitate a better understanding of factors influencing the choice of preventive and control measures taken by men and women, with the goal of supporting veterinary extension and policy makers to design effective disease control strategies for PPR and RVF in different production systems.

This study had a few limitations. First, the study sites were purposively chosen based on the reported occurrence of PPR and RVF, which might have introduced confounding factors. Qualitative interviews were conducted within a few selected

sub-counties; therefore, they may not be generalizable to other areas. However, the repetitions of narratives and illustrations from the discussants under each thematic area were adequate to highlight experiences from men and women with similar backgrounds and therefore similar influencing factors. Given the low awareness of RVF and PPR, recommendations about prevention and control were inferred from how people prevented or controlled other ruminant diseases. The small sample size of female animal health workers was a limitation but also reflected the reality in Uganda that few women, or none in some of the districts, had ventured into the veterinary profession, making it difficult to have a good representation of women as KIIs. While the collected demographic information revealed diverse age groups (above 18 years), we did not intentionally target youth categories. More focus on youth would improve future studies. Although we investigated the roles of men and women, we did not intentionally target children as key informants because of ethical concerns even though their roles in ruminant disease management are crucial in these production systems. The assumption was that adults would be able to report on their children's roles. GAF structure includes the elderly for consideration as a separate category but there were not enough elderly respondents for separate analysis. Lastly, the co-authors of this paper acknowledge that their background could have influenced the research processes given that in qualitative research the researcher's background and biases influence the study [94]. PL, DT, BB, HK and KR are veterinarians with research interests in vaccination and epidemiology of different livestock diseases under different production systems including participatory epidemiology: ZAC, JN, MA, EO, and PL are social scientists specializing in the intersection between gender and animal health.

## Conclusions and recommendations

The use of the GAF and SEM has facilitated a better understanding of the influencing factors at multiple levels, demonstrating the need to engage multiple stakeholders at different levels: policy makers, veterinary extensionists, research institutions, development partners and the private sector. At the policy level, it would be worthwhile to address issues related to access to vaccination and support targeted policies and programs to ensure fair distribution and support for all livestock owners regardless of herd size or gender. At the level of veterinary extension, it would be important to identify avenues which are socially acceptable and effective for reaching smallholder livestock keepers, especially women who own livestock, and might be potential beneficiaries of such information. Ensuring timely and accurate information dissemination and providing targeted support to women and other vulnerable livestock keepers would facilitate access to veterinary services. Assessing training needs for men and women livestock keepers related to disease management could be explored before designing tailored training programs to enhance women's and men's skills in identifying and managing livestock health related issues. This would provide gender specific information on preferred location, timing and training content.

Some ongoing development and research projects are working to increase women's access to livestock vaccines by pairing technical interventions, such as those described above, with community activities and dialogues that directly challenge restrictive gender norms [95]. While identifying restrictive gender norms was beyond the scope of the study, some of the findings hint at norms that restrict women from accessing these vaccines. For example, the norm that livestock keeping is only for men could mean women are not seen as livestock keepers by organizers of a training and are therefore left out. The norm that women shouldn't have close personal relationships with a man who is not her husband or a relative could prevent her from interacting with a male animal health worker, something that men said, "gives them experience in the treatment of animal diseases." Acknowledging and addressing the role of gender norms in vaccine access is recommended for future research and development activities.

Lastly, a topic for future research is investigating how information networks within production systems could support more women to make decisions about livestock disease control. For example, how can disease reporting networks within households and communities enhance livestock disease prevention and control? Which networks are most preferred? By whom, when, and why? These questions combined with a gender lens to acknowledge the power hierarchies of the way information moves could be used to inform more equitable and effective information campaigns for vaccination of livestock against PPR, RVF, and other diseases.

## Supporting information

**S1 File. Focus group discussion guide.**
(DOCX)

**S2 File. Key informant interview guide.**
(DOCX)

**S3 File. Coding framework used in NVivo.**
(DOCX)

**S1 Table. Analysis of factors influencing disease control options uptake by gender and production system.**
(DOCX)

## Acknowledgments

We acknowledge the contributions of our implementing partners. The authors thank Pamela Wairagala for her valuable support in editing the manuscript, livestock keepers who participated in this study, as well as the district officials for authorizing the data. Thank you to Paul Lumu from the Ministry of Agriculture, Animal Industry and Fisheries (MAAIF) for his support as the PPR representative for the Boosting Uganda's Investment in Livestock Development (BUILD) project.

## Author contributions

**Conceptualization:** Jane Namatovu, Peter Lule, Marsy Asindu, Zoë A. Campbell, Dan Tumusiime, Emily Ouma.

**Data curation:** Jane Namatovu, Peter Lule, Marsy Asindu.

**Formal analysis:** Jane Namatovu, Peter Lule, Marsy Asindu, Zoë A. Campbell.

**Funding acquisition:** Henry Kiara, Bernard Bett, Kristina Roesel.

**Investigation:** Jane Namatovu, Peter Lule, Marsy Asindu, Dan Tumusiime.

**Methodology:** Jane Namatovu, Peter Lule, Marsy Asindu, Zoë A. Campbell, Dan Tumusiime, Emily Ouma.

**Project administration:** Dan Tumusiime, Henry Kiara, Bernard Bett, Kristina Roesel, Emily Ouma.

**Resources:** Henry Kiara, Bernard Bett, Kristina Roesel, Emily Ouma.

**Software:** Bernard Bett, Kristina Roesel.

**Supervision:** Zoë A. Campbell, Henry Kiara, Bernard Bett, Kristina Roesel, Emily Ouma.

**Validation:** Jane Namatovu, Peter Lule, Marsy Asindu, Zoë A. Campbell, Dan Tumusiime, Henry Kiara, Bernard Bett, Kristina Roesel, Emily Ouma.

**Visualization:** Jane Namatovu, Peter Lule, Marsy Asindu, Zoë A. Campbell.

**Writing – original draft:** Jane Namatovu, Peter Lule, Zoë A. Campbell.

**Writing – review & editing:** Jane Namatovu, Peter Lule, Marsy Asindu, Zoë A. Campbell, Henry Kiara, Bernard Bett, Kristina Roesel, Emily Ouma.

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
