## [Decision Letter · Decision Letter 0]

29 Feb 2024

PONE-D-24-01412Gender roles in ruminant disease management in Uganda: Implications for the control of Peste des petits ruminants and Rift Valley fever.PLOS ONE

Dear Dr. Jane,

Thank you for submitting your manuscript to PLOS ONE. After careful consideration, we feel that it has merit but does not fully meet PLOS ONE’s publication criteria as it currently stands. Therefore, we invite you to submit a revised version of the manuscript that addresses the points raised during the review process.

**ACADEMIC EDITOR: ** After revising your manuscript, the reviewer's advice to perform minor revisions for improvement of the manuscripts, the reviewer's comments are attached below.

We look forward to receiving your revised manuscript.

Kind regards,

Nussieba A. Osman, Dr. Med. Vet.

Academic Editor

PLOS ONE

Journal Requirements:

“This work was supported by the German Federal Ministry of Economic Cooperation and Development (BMZ) through the project Boosting Uganda’s investment in livestock development (BUILD) (Grant

number BMZ001). Additional support was received from the CGIAR Research Programs on Livestock and Agriculture for Nutrition and Health. We also acknowledge the CGIAR Fund Donors”

Reviewers' comments:

Reviewer's Responses to Questions

**Comments to the Author**

1. Is the manuscript technically sound, and do the data support the conclusions?

Reviewer #1: Yes

Reviewer #2: Yes

2. Has the statistical analysis been performed appropriately and rigorously? 

Reviewer #1: N/A

Reviewer #2: Yes

3. Have the authors made all data underlying the findings in their manuscript fully available?

Reviewer #1: Yes

Reviewer #2: Yes

4. Is the manuscript presented in an intelligible fashion and written in standard English?

Reviewer #1: Yes

Reviewer #2: Yes

5. Review Comments to the Author

Reviewer #1: Dear Jane et al.,

I enjoyed reading the manuscript PONE-D-24-01412 entitled “Gender Roles in Ruminant Disease Management in Uganda: Implications for the Control of Peste des Petits Ruminants and Rift Valley Fever.”

I got the opportunity to expand my understanding and knowledge of livestock disease control and prevention measures in Uganda. I have extended experience with animal health efforts toward disease prevention and control in eastern and western Africa but haven’t directly worked in the specific context of Uganda. So, I tried to be generic to some extent, acknowledging that aspect yet being very specific and thorough regarding scientific expectations from such an article. So below, you will find two sections, mainly a brief takeaway of my understanding of the study and a comments/question section ranging from minor to major. I remain available to expand further if some of my comments are not well understood.

Summary

The study addressed the following two research questions:

What do men and women do to prevent or control livestock diseases that affect them and their livestock?

What factors influence the choice of the disease control measures taken?

It utilized a social-ecological model (SEM) and gender-analysis framework to examine factors influencing disease control options for men and women in three production systems for two livestock diseases in Uganda: Peste des petits ruminants (PPR) and Rift Valley fever (RVF).

Some of the main results are that disease identification was majorly inclined towards gender roles, with men identifying diseases based on their perceptions and experiences during the post-mortem of dead animals and women during livestock management.

Men and women make different management decisions to prevent or control livestock diseases. The choices are often influenced by the availability of resources such as money, time, labor, information, the cost of technology, and hierarchical power relationships within the household/community.

Comments and questions

The reviewer appreciates the strong literature review provided and the efforts of the authors to detail the sampling process.

1. The sites were selected on purpose based on epi reports in the areas. I suggest adding a section about the seroprevalence, vaccination coverage, and post-vaccination immunity levels for each disease and, if possible, per species (goats, sheep, cattle...) to make a case for the selections of those sites. Ideally, a table presenting those statistics nationally would easily justify the selection of sites on “purposes” unless I failed to understand the purposes mentioned.

2. Decision vs. Action/Operation is not just based on the household head. The literature on intrahousehold studies made several findings about who does what in the household regarding (decisions and actions) regarding husbandry, healthcare, markets/sales, and income spending. Drawing a list based on gender and household head might be a little too simplistic, given the objectives of the study. Some activities are not done by the women or men head of a household or rather by the young girl or boy in the household. It would have been more insightful to have them select activities and decisions they take and do in their household to ensure that the right person is in the room (FGD).

3. Provide a section of the SEM and show how the model has helped answer the research questions. The methods sections did not specifically explain how the SEM was built and used. Instead, all the results seem mainly based on FGD and KIIs, whose concrete findings are also barely mentioned. Hence, one can wonder if using FGD, KII, and participatory appraisals wouldn’t be enough to do such an analysis. Or what is the added value of the SEM?

4. One of the major animal health constraints in disease control is mobility data. In pastoral production systems, animals are rarely kept in fences, so mobility creates contacts at grazing, water points, and markets. The reviewer could not find real pieces of evidence in the articles on how these critical elements of disease control and prevention were handled.

5. At the end of the participatory sections, were there any restitutions, lessons learned, or best practices for disease prevention and control to the participant? What have they learned differently from their initial knowledge? One of the results of the analysis was that women could have been misinformed or lacked access to the right information. So, the reviewer would expect a restitution or validation meeting during which authors or health workers have updated their knowledge to impact them.

6. One aspect of disease prevention the reviewer expected was on herder’s willingness to vaccinate. The reviewer appreciates the various elements in Fig 2, especially those on perceptions of vaccines (trust, availability, side effects) and WTV for vaccination. While the availability of vaccines and other drugs at the different shops is limited due to technical and infrastructural issues, it would have been an added value to inquire if money or the cost of vaccines represents a constraint for vaccination either during or outside of national campaigns. Usually, vaccination is subsidized during campaigns to incentivize more farmers to afford the vaccination cost and boost immunity. Adding these elements could reinforce some of your conclusions.

7. The reviewer suggests adding a section for the conclusion as the authors did for the introduction. It is also important to make a clear differentiation between discussion and recommendation. Mixing the two has made that section overly long. Discuss only the study results first and add a couple of paragraphs in which you might provide your recommendations.

8. Lines 98: Reference with author name is provided in the text, unlike others. Kindly harmonize the references across the document. It may also apply to lines 643 and 634, though I am unsure. It is good to review this one more time throughout the document.

9. Another point that stood out is the gender ratio for the KIIs. How sound is it for a gender-specific study that is trying to understand gender differences in disease control and prevention to interview 32 persons with only 6 women? This would have been statistically wrong. To some extent, by increasing the sample size, one would also expect to increase the chances for various experiences and knowledge shared of the subject.

10. In Fig 2 and other parts (Table 2), where numbers are provided, it would be good to explain how those numbers are generated. You have multiple FGDs; each could be used to generate these tables or Figs. How did you end up with one table and figure? Kindly expand the methods sections with an explanation of these questions.

11. Your results are not presented per site or district but per animal production system… the reviewer is convinced that there isn’t a homogeneity of results across all sites. Hence, discussing the specificities of some of these sites could shed light on where or locations that need more attention and investments.

Good luck.

Reviewer.

Reviewer #2: 1. the title of the paper is catch, and concise with the content of the paper

2. Abstract is relatively well written, it is with acceptable length and contain all mandatory information

3. The introduction subsection is well written, it gives an overview of the key issues contained in the paper

4. The theoretical orientation and analysis for the paper is well presented and they are relevant to the paper

5 The research questions are in line with the objectives of the paper and they add clarity in the causal effect relationship of the key study variables

6. the methodology subsection is well written, the research design, sampling procedures and data analysis plans are relevant and adequate. However, despite the fact that the study was qualitative it is important also to acknowledge that quantitative data were also collected as it has been presented in Table 1 page 11

7. The is a narrow use of the term gender to only refer to men and women, unless otherwise operationally defined, table 2 may require revisiting to include other gender categories e.g. youth male and female

8. The presentation of the findings as it appear on page 13 -27 require revisiting by improving the citation of the qualitative data, the subsection should clearly show whether the findings are from FGD or KI, location and authority of the one giving such information in terms of the KIIs

9. The discussion of the findings requires some improvements, it should reduce the repetition of findings rather, synthesis it to generate conclusions in view of the key findings, the subsection also have minimal use of literature. After generating inferences from the finding the author should also show whether his/her finding compare with others and give justification why ?

10. the conclusions and recommendations are not glaring in the discussion subsection, it should be overhauled by either showing clear conclusion and recommendations within the same subsections or the new subsections on conclusions and recommendations be added

6. PLOS authors have the option to publish the peer review history of their article (what does this mean? ). If published, this will include your full peer review and any attached files.

**Do you want your identity to be public for this peer review?** For information about this choice, including consent withdrawal, please see our Privacy Policy .

Reviewer #1: No

Reviewer #2: No

---

## [Author Response · Author response to Decision Letter 1]

28 May 2024

April 12, 2024

Dear Dr Nussieba A. Osman,

We would like to thank you for giving us the opportunity to submit a revised draft of the manuscript “Gender roles in ruminant disease management in Uganda: Implications for the control of peste des petits ruminants and Rift Valley fever. We appreciate the time and effort that you and the reviewers dedicated to providing feedback on our manuscript. Our responses to the reviewers’ comments are in italics below and the changes are highlighted within the revised manuscript. Where line numbers are indicated, they refer to the revised manuscript (Track changes mode).

Comment 1:

Response: Thank you for pointing this out. The file naming has been revised as per the PLOS ONE’s style requirements.

Comment 2:

Thank you for stating in your Funding Statement: “This work was supported by the German Federal Ministry of Economic Cooperation and Development (BMZ) through the project Boosting Uganda’s investment in livestock development (BUILD) (Grant number BMZ001). Additional support was received from the CGIAR Research Programs on Livestock and Agriculture for Nutrition and Health. We also acknowledge the CGIAR Fund Donors”. Please provide an amended statement that declares *all* the funding or sources of support (whether external or internal to your organization) received during this study, as detailed online in our guide for authors at http://journals.plos.org/plosone/s/submit-now. Please also include the statement “There was no additional external funding received for this study.” in your updated Funding Statement. Please include your amended Funding Statement within your cover letter. We will change the online submission form on your behalf.

Response: Thank you for highlighting this. We have now included all sources of support and added a statement that there are no other additional external funding sources:

This work was supported by the German Federal Ministry of Economic Cooperation and Development (BMZ) through the project Boosting Uganda’s investment in livestock development (BUILD) (Grant number BMZ001) and One Health Research, Education and Outreach Centre in Africa (OHRECA) (Grant number BMZ002). Additional time support was received from the CGIAR Initiative Sustainable Animal Productivity (SAPLING) which is supported by the contributors to the CGIAR Trust Fund. (https://www.cgiar.org/funders). There was no other additional external funding received for this study.

Comment 3:

We note that Figure 1 in your submission contain [map/satellite] images which may be copyrighted. All PLOS content is published under the Creative Commons Attribution License (CC BY 4.0), which means that the manuscript, images, and Supporting Information files will be freely available online, and any third party is permitted to access, download, copy, distribute, and use these materials in any way, even commercially, with proper attribution. For these reasons, we cannot publish previously copyrighted maps or satellite images created using proprietary data, such as Google software (Google Maps, Street View, and Earth). For more information, see our copyright guidelines: http://journals.plos.org/plosone/s/licenses-and-copyright.

a. You may seek permission from the original copyright holder of Figure 1 to publish the content specifically under the CC BY 4.0 license. We recommend that you contact the original copyright holder with the Content Permission Form (http://journals.plos.org/plosone/s/file?id=7c09/content-permission-form.pdf) and the following text: “I request permission for the open-access journal PLOS ONE to publish XXX under the Creative Commons Attribution License (CCAL) CC BY 4.0 (http://creativecommons.org/licenses/by/4.0/). Please be aware that this license allows unrestricted use and distribution, even commercially, by third parties. Please reply and provide explicit written permission to publish XXX under a CC BY license and complete the attached form.” Please upload the completed Content Permission Form or other proof of granted permissions as an "Other" file with your submission.

b. If you are unable to obtain permission from the original copyright holder to publish these figures under the CC BY 4.0 license or if the copyright holder’s requirements are incompatible with the CC BY 4.0 license, please either i) remove the figure or ii) supply a replacement figure that complies with the CC BY 4.0 license. Please check copyright information on all replacement figures and update the figure caption with source information. If applicable, please specify in the figure caption text when a figure is similar but not identical to the original image and is therefore for illustrative purposes only. The following resources for replacing copyrighted map figures may be helpful: USGS National Map Viewer (public domain): http://viewer.nationalmap.gov/viewer/

Response: Thank you again for highlighting the improvements we need to make to our manuscript. The map was generated by one of the co-authors, Marsy Asindu. Shapefiles for this map were adopted by the authors from the Ugandan Energy Sector GIS Working Group Open Data Site using ArcGIS: http://data-energy-gis.opendata.arcgis.com/

The caption for fig 1 has been added as follows: Fig 1: Map showing the study districts within Uganda. (Original shape files for the map were adopted by the authors with ArcGIS from the Ugandan Energy Sector GIS Working Group Open Data Site: http://data-energy-gis.opendata.arcgis.com/)

Comment 4

Please review your reference list to ensure that it is complete and correct. If you have cited papers that have been retracted, please include the rationale for doing so in the manuscript text or remove these references and replace them with relevant current references. Any changes to the reference list should be mentioned in the rebuttal letter that accompanies your revised manuscript. If you need to cite a retracted article, indicate the article’s retracted status in the References list and also include a citation and full reference for the retraction notice.

Response: Thank you for pointing this out. We have only added other references where needed but not removed as indicated below.

Reference # Reference

Lines

47 Nkamwesiga et al. (2023) 150

48 Nkamwesiga et al. (2020) 150

49 Nkamwesiga et al. (2019) 150

50 Ruget et al. (2019) 150

51 Tumusiime et al. (2019) 150

52 Tumusiime et al. (2023) 150

53 Nyakarahuka et al. (2016) 151

58 Akwongo et al. (2022) 689

59 Ruhweza et al. (2010) 690

60 Gitonga et al. (2016) 690

62 Nanfuka et al. (2018) 701

66 Ilukor et al. (2012) 883

Response to reviewers’ comments: Reviewer #1

The reviewer appreciates the strong literature review provided and the efforts of the authors to detail the sampling process. The sites were selected on purpose based on epi reports in the areas. I suggest adding a section about the seroprevalence, vaccination coverage, and post-vaccination immunity levels for each disease and, if possible, per species (goats, sheep, cattle...) to make a case for the selections of those sites. Ideally, a table presenting those statistics nationally would easily justify the selection of sites on “purposes” unless I failed to understand the purposes mentioned.

Response: We appreciate this valuable suggestion and thank you for pointing out the misunderstanding. The selection of the sites was based on epidemiology reports and livestock densities in those areas as highlighted in the paragraph starting from line 143 (Description of study sites), including references.

Decision vs. Action/Operation is not just based on the household head. The literature on intrahousehold studies made several findings about who does what in the household regarding (decisions and actions) regarding husbandry, healthcare, markets/sales, and income spending. Drawing a list based on gender and household head might be a little too simplistic, given the objectives of the study. Some activities are not done by the women or men head of a household or rather by the young girl or boy in the household. It would have been more insightful to have them select activities and decisions they take and do in their household to ensure that the right person is in the room (FGD).

Response: We agree with the reviewer’s assessment. Although the selection criteria for the FGD participants was based on the household head, the Focus Group Discussion tool was designed to tease out the gender roles for other household members such as girls/boys who could not be interviewed based on ethical issues. The assumption was that the selected adults would correctly give information about the roles/responsibilities of other household members, for example girl/boys. This information has been captured within the results as follows:

lines 327-328: The role of boys in ruminant disease management mostly in pastoral areas was limited to grazing, identifying unhealthy animals while out grazing and reporting to parents or guardians.

Lines 424426: Boys under the instruction of their fathers checked on grazing areas and watering points before taking the animals there.

lines 324-325: Girls supported their mothers to clean livestock shelters.

Lines 364-365: Both girls and boys handpicked ticks as part of the vector control measure.

Provide a section of the SEM and show how the model has helped answer the research questions. The methods sections did not specifically explain how the SEM was built and used. Instead, all the results seem mainly based on FGD and KIIs, whose concrete findings are also barely mentioned. Hence, one can wonder if using FGD, KII, and participatory appraisals wouldn’t be enough to do such an analysis. Or what is the added value of the SEM?

Response: Thank you for pointing this out. We have made the suggested changes, and these are highlighted in lines 169-178: The intrapersonal level explored attitudes, beliefs, perceptions and knowledge of PPR/RVF. The interpersonal level examined the social influences ranging from interactions with animal health workers, fellow livestock keepers, social functions in the community, family responsibilities and aspects of decision making. The community level explored veterinary/vaccination information outreach and feeding practices at community level, perception of livestock disease risk and outbreaks, while the institutional level considered vet service costs, distance to service centers, availability of animal health providers, information access and social economic inequalities between livestock keepers. The policy level explored rules and regulations regarding PPR/RVF prevention and control.

One of the major animal health constraints in disease control is mobility data. In pastoral production systems, animals are rarely kept in fences, so mobility creates contacts at grazing, water points, and markets. The reviewer could not find real pieces of evidence in the articles on how these critical elements of disease control and prevention were handled.

Response: This has been highlighted in lines 314-316 where in pastoral system-women identified and isolated sick animals in the morning before they are taken out for grazing.

Lines 424-426: Also, boys under the instruction of their fathers inspect water points before taking the animals for watering.

Lines 427-429: Goats and sheep which are identified as sick are tethered separately to avoid mixing them with the healthy ones or separated them in an isolated kraal until they are vaccinated or treated then released later for grazing.

At the end of the participatory sections, were there any restitutions, lessons learned, or best practices for disease prevention and control to the participant? What have they learned differently from their initial knowledge? One of the results of the analysis was that women could have been misinformed or lacked access to the right information.

Response: This is a great suggestion. The information generated from this study was used by partner organization mandated with veterinary extension service (VSF-G) to design awareness and sensitization campaigns for livestock keepers and veterinary extension workers on disease identification and preventive measures. Additionally, the key outcomes of this study have been summarized in a policy brief and shared with relevant authorities in the Ministry of Agriculture, Animal Industry and Fisheries (MAAIF) who are set to further incorporate it as part of their extension action point. Finally, the government developed a national PPR strategy document and even mobilized more than 1.5m doses of vaccines, not only due to this study but due to the evidence from the overall project.

One aspect of disease prevention the reviewer expected was on herder’s willingness to vaccinate. The reviewer appreciates the various elements in Fig 2, especially those on perceptions of vaccines (trust, availability, side effects) and WTV for vaccination. While the availability of vaccines and other drugs at the different shops is limited due to technical and infrastructural issues, it would have been an added value to inquire if money or the cost of vaccines represents a constraint for vaccination either during or outside of national campaigns.

Response: The authors appreciate this suggestion. However, since this study was part of a wider project (Boosting Uganda’s investment in Livestock Development), other social economic aspects such as willingness to pay for PPR and RVF vaccines were considered in other social economic research studies on PPR and RVF which are in the pipeline for publication. However, RVF vaccine is not licensed for use in Uganda and PPR vaccines is not subsidized in Uganda. While the government procures the doses, the farmer must pay for the vaccine and the service (transport and administration through the government veterinarians). We have added some of these elements suggested to the conclusions (lines 761ff).

The reviewer suggests adding a section for the conclusion as the authors did for the introduction. It is also important to make a clear differentiation between discussion and recommendation. Mixing the two has made that section overly long. Discuss only the study results first and add a couple of paragraphs in which you might provide your recommendations.

Response: We appreciate this suggestion. We have added a section on conclusions and recommendations and a separate one on discussion.

Lines 98: Reference with author name is provided in the text, unlike others. Kindly harmonize the references across the document. It may also apply to lines 643 and 634, though I am unsure. It is good to review this one more time throughout the document.

Response: Thank you for highlighting this. We have reviewed the references across the entire document. Corrections have been made to lines 97,723, 753 and 774-777.

Another point that stood out is the gender ratio for t

---

## [Decision Letter · Decision Letter 1]

2 Aug 2024

PONE-D-24-01412R1Gender roles in ruminant disease management in Uganda: Implications for the control of peste des petits ruminants and Rift Valley fever.PLOS ONE

Dear Dr. Jane,

Thank you for submitting your manuscript to PLOS ONE. After careful consideration, we feel that it has merit but does not fully meet PLOS ONE’s publication criteria as it currently stands. Therefore, we invite you to submit a revised version of the manuscript that addresses the points raised during the review process.

**ACADEMIC EDITOR: **

The manuscript needs major revision in order to be accepted for publications in PLOS ONE. Please consider revise your manuscript considering all points raised by the reviewers.

We look forward to receiving your revised manuscript.

Kind regards,

Nussieba A. Osman, Dr. Med. Vet.

Academic Editor

PLOS ONE

Reviewers' comments:

Reviewer's Responses to Questions

**Comments to the Author**

1. If the authors have adequately addressed your comments raised in a previous round of review and you feel that this manuscript is now acceptable for publication, you may indicate that here to bypass the “Comments to the Author” section, enter your conflict of interest statement in the “Confidential to Editor” section, and submit your "Accept" recommendation.

Reviewer #1: All comments have been addressed

Reviewer #3: (No Response)

2. Is the manuscript technically sound, and do the data support the conclusions?

Reviewer #1: Yes

Reviewer #3: No

3. Has the statistical analysis been performed appropriately and rigorously? 

Reviewer #1: Yes

Reviewer #3: N/A

4. Have the authors made all data underlying the findings in their manuscript fully available?

Reviewer #1: Yes

Reviewer #3: Yes

5. Is the manuscript presented in an intelligible fashion and written in standard English?

Reviewer #1: Yes

Reviewer #3: No

6. Review Comments to the Author

Reviewer #1: Dear Jane et al.,

Let me start by thanking you and the rest of your team for the improved manuscript quality. Overall, you found my comments and questions insightful. I appreciate your positive feedback. I have recommended accepting the revised version of the manuscript.

However, it would be best for you and your team to proofread the document for additional typos here and there. Besides, I suggest adding a line or two to account for the study's limitations and list all assumptions made. I still have some reservations that a household head who is usually more on the management and decision-making side would know enough about other tasks completed that are often overlooked by other members who are only sometimes underage (responding to your ethical comment).

The changes made to the suggested tables and figures and the additional sections for the conclusion and recommendations are all well-suited. The reviewer genuinely appreciates the upcoming extension efforts to raise awareness and prompt behavioral changes through a partnership with the veterinary extension service (VSF-G) and the Ministry of Agriculture, Animal Industry and Fisheries (MAAIF).

Well done, and good luck.

Reviewer #3: I read the clear version of the revised manuscript appearing first in the pdf, line numbering refers to that version.

General comments

The introduction introduces the subject, the local setting concerning PPR and RVF, the research questions and the methodologies to be used in a very nice way. Unfortunately, the rest of the manuscript is not of the same quality and does not deliver what is promised in the introduction.

The description of the methodology, especially data analysis, lacks important clarity and details, see the detailed comments below. Among the missing details not mentioned in the comments below is a section on the researchers’ positionality. As deductive coding is used it is important to reflect and mention the researchers’ previous knowledge and experiences to explain how the analysis process was driven by the researchers’ position. It is further not clear from either the methodology or the results how the SEM and the GAF was used, it now appears that the SEM was only use was to divide the factors influencing disease control decision in the five mentioned categories and there is no description of GAF-analysis in the methodology at all. Moreover, it is not clear how the deductive codes described in the code books (S1 and S2) was used in the analysis to give the subsections described in the results, or if these subsections are indeed what is described as themes/common themes/analytical themes. This lack in the description of the methodology and the analysis makes it impossible to judge the methodological rigour, and if the conclusions are based on the data. I have further have some major concerns about the FGDs, see my comment on S1 below.

All through the results section it needs to be made clearer if the reported results are valid for all systems, a specific system, or only one specific district.

The discussion lacks depth and the discussion of the findings is not on the level of abstraction and reflection that is expected from a qualitative study in this journal.

The text would further highly benefit from professional language editing.

Detailed comments:

Supplementary material 1 (FGD guide): The FGD-guide includes very many and quite detailed questions, even if only considering section A. With section B and C the content becomes enormous. It is written in the S1-file that different FGDs should be used for section A versus sections B and C, but this is not reflected in the manuscript, was those sections used for another study? Please clarify. I also find it unusual to have so many and detailed questions in a FGD in which one normally aims at having a free and open discussion driven by the participants. With so many questions the guide looks more like a semi-structured protocol for an individual interview. The length/content of the guide further raises several questions: How long time did the FGDs take? How was participant engagement ensured during the entire process? How was the results used, as for example the results from the PE are not included in this manuscript? For question 3a the probing questions and the instruction to the facilitator don’t seem to match (common diseases/all diseases/description of the diseases/zoonotic potential vs ranking based on highest mortality). Were the follow up questions (which are very many) in questions 4-6 asked for all listed diseases? If not, for which ones and how was the selection agreed on?

Supplementary material 2 (KII guide): as for the FGD-guide, this guide also contains very many (interesting and important) questions. But the time given is only 25 minutes – could really all questions be answered in such a short time? And it further seems like a waist not to use the opportunity to let the conversation cover the questions more in depth rather than rushing through to finish in 25 minutes?

Line 30: in the abstract you might want to skip mentioning the name of the software used.

Line 46-49: the sentence is very long, please shorten it or divide it in two.

Line 62: I presume the “in” should be placed after “children”.

Line 61-62: milking is an important activity that should be mentioned here.

Line 78: the abbreviation (SR) not previously introduced.

Line 87: please remove “as an asset”, or explain why these words are needed here.

Line 89: please remove “chances of”.

Line 90: either add a comma after “new” or remove the comma before “and”.

Line 91: instead of “burning and burying” I would prefer “safe disposal of animals that have died from diseases” as there are several ways to do this without spreading disease.

Line 93: for RVF, safe handling of abortions (which might present without the animal being “sick)” is also a very important preventive measure, please add this.

Line 93: you need to mention the vaccination guidelines also for PPR.

Line 95-107: this seems not to belong to the subsection PPR and RVF in Uganda but rather to the previous, please move and integrate to make sure to avid overlaps.

Line 108-132: his also seems not to belong to the subsection PPR and RVF in Uganda but rather to a subsection on gender in decision making and/or decision making determinant models. Please move/integrate.

Line 116: which are these five key determinants? Are they only/exactly five? Please clarify.

Line 127: what is meant with “all-round” in this context?

Line 128: more realistic than what? Please clarify.

Line 133: this seems to be the beginning of another subsection? Please clarify. It would further be a good place to introduce the objectives of the study.

Line 144-147: a blank is missing or is in surplus in a couple of places in this sentence, please check and correct.

Line 150: please define pastoral/agropastoral.

Line 150-151: does this refer to all study districts? If so, please move the sentence to reflect that, and if not, please clarify which districts.

Line 153: do you mean disease transmission or disease prevalence here?

Line 191-193: the description of the GAHW and CAHW-workforce here seems misplaced, were CAHWs included as key-informants? Please clarify.

Line 205: according to the FGD-guide in S1 the PE-tools were not a complement, but an integrated part of the FGDs? Please clarify and report of the results from the PE-exercises.

Line 207: you need to add some more information on facilitation, language and translation. Who in the research team facilitated the interviews (FGDs and KIIs)? In what language(s) was the FGDs and KIIs done? By whom in the research team? Was notes taken in English? If not – were the note translated? How/by whom in the research team translated the interview guides? What do you mean by “in person”?

Line 219: are the “team members” co-authours? Please clarify.

Line 223: please mention the research questions.

Line 224-231: please clarify the difference between/define what you mean by: theme, topic, common theme, analytical theme, identified theme, level of interest, subtheme.

Line 221-232: please clarify the analysis-process. Did you really code before uploading the data in Nvivo? Did you use deductive coding or were themes emerging from the data? It seems (but this is not clearly described) that deductive coding was used, with the codes used in the respective code book for RVF and PPR), this needs to be described in more details, and in my opinion the codes should be included in the manuscript (not as supplementary material).

Line 228: please clarify what you mean by “asses” in this regard (number of, presumed efficiency, quality, cost-benefit etc).

Line 236: please remove the space in subtheme.

Line 254: see my comments on the analysis and the mentioning of different kinds of themes. It would be good to start the results section by mentioning the themes that arose (emerged) during the analysis (if any).

Line 255-258: this belongs in the methods-section. In addition, it is stated here that the factors were organized according to the SEM, which is what I commented on in the general comment, did the SEM include some more analytical elements, or only to organize the factors according to these pre-defined categories?

Line 256: I presume the “socio-ecological framework” is the same as the SEM? Please clarify.

Table 1 and 2: please adjust the formatting. Regarding the note for Table 1: I don’t see any SDs in this table? Foot notes are normally used for this kind of information.

Table 2 and Figure 2: does “mixed” refer to “mixed crop-livestock system” or a mix of pastoral/agropastoral systems? Please clarify.

Line 268-69: a verb seems to be missing in the first part of the sentence.

Line 282: please check the level of the subsection titles to make the reading easier and clearer.

Line 294-295: please check the sentence for grammar and synthax.

Line 295-296: please check the sentence for grammar and synthax.

Line 297: is this statement valid for all agropastoral study areas or only for Isingoro? Please clarify.

Line 302-317: this seems not to fit under prophylactic measures, please move to another subsection.

Line 303: was this only done in “sudden deaths”? Or do you mean in all cases of animals dying (by themselves) without known cause of death?

Line 303-304: please consider the wording “local post mortem”, I think maybe something like “carcass opening” would be a more correct description?

Line 305-307: was the meat distributed and the skin used as described only if the animal had died a “sudden death”? Please clarify.

Line 308: quotes are normally written in italics.

Line 308-09 and 312-14: these 2 quotes seem to illustrate the same thing, please remove one.

Line 315-317: which result do you want to illustrate with this quote? Please clarify.

Line 317: a full stop is missing at the end of the sentence.

Line 322: the purpose of this statement is not clear. Was spraying done more frequently (compared to pastoral or mixed systems?) in agropastoral systems, or was it done more as a family activity (compared to what/where?). Please clarify.

Line 328: should it be ”kids and lambs” or were only goats kept. Please clarify.

Line 355-356: the purpose of this statement is not clear. Please clarify.

Line 365: the mentioning of “from the FGDs” are puzzling as it seemed to me that all results reported so far were from the FGDs (compare line 256). Please clarify.

Line 368-369: please remove “on the farm”.

Line 377, 381, 396: please check for typos.

Line 385: what is a “personal farm” and what do you mean by “shifting” in this sentence? Please clarify.

Line 387: how was movement limited by women? Please clarify.

Line 392: at what occasion were animals moved? Please clarify.

Line 399-403: this sentence seems to be related to the one on line 385? Please put them together and avoid repeating/overlapping.

Line 410-413: please check the sentence for grammar and syntax.

Line 443: it needs to be described in the methodology section how this order of importance established. Please clarify.

Line 485: please introduce the quote with a full sentence.

Line 507-509: please check the sentence for grammar and syntax.

Line 655-656: I failed to see description of the gender analysis framework in methodology section, or the results coming from this method. Please clarify.

Line 659-661: this seems to be results that are not reported in the results-section. Please don’t introduce new results in the discussion-section.

Line 661-663: I don’t understand this sentence, please try to clarify/rewrite.

Lune 663-666: idem. Plus the study does not seem to be designed to assess risk-factors so I don’t understand how the statement is based on the results?

Line 667: please clarify what “more” refers to (more than what)?

Line 669; a blank is missing before the bracket.

Line 678 and in other places: I think “self-medication” is not the best term for what you want to describe, for me “self-medication” sounds like someone is medicating him/herself. Please look for another way to describe the phenomenon of medicating ones animals without consulting an animal health professional .

Line 689-690: please check the sentence for syntax.

Line 690-692: I confess to not having read the article in the reference, but disregarding what it says, the association between purchasing acaricides and domestic violence and in the next step suicide needs to be more thoroughly introduced here.

Line 694-696: please check the sentence for grammar and syntax.

Line 698: I presume you mean “more common”? Please check.

Line 698-670: I don’t understand this sentence, please try to clarify/rewrite.

Line 713-714: this seems to be a conclusion or possibly a recommendation. Please move to a more appropriate section.

Line 726-728: please check the sentence for syntax.

Line 730: a blank is missing between the two sentences.

Line 749: please change from capital to nominal i in insights.

Line 759: there seems to be a surplus full stop, please remove one.

Line 766: please check for typos.

Line 781-798: this seems not to be conclusions or recommendations, please move to a more appropriate section.

Line 782: I presume you mean “reported occurrence”?

Line 791-793: please check the sentence for syntax.

Line 1067: please check the numbering of the supplementary material files.

Fig 1: I presume that the “corridor region” in the legend refers to the cattle corridor? I further don’t see any reference to the cattle corridor in the text (but I might have missed that), please remove the cattle corridor from the map or mention it in the description of the study area(s).

7. PLOS authors have the option to publish the peer review history of their article (what does this mean? ). If published, this will include your full peer review and any attached files.

**Do you want your identity to be public for this peer review?** For information about this choice, including consent withdrawal, please see our Privacy Policy .

Reviewer #1: No

Reviewer #3: No

---

## [Author Response · Author response to Decision Letter 2]

1 Oct 2024

Dr. Nussieba A. Osman

Academic Editor

PLOS ONE

September 26, 2024

Re: Revision required for manuscript PONE-D-24-01412R1

Dear Sir,

We appreciate your feedback and the opportunity to resubmit a revised manuscript. We also appreciate the reviewers for taking the time to carefully review the manuscript and give detailed and constructive comments. We have incorporated the changes reflecting all the suggestions provided by the editor and reviewers. For ease of reading, we have grouped our responses by section.

Below is our point-by-point response to each comment. If changes were made in the revised manuscript we indicate line numbers referring to the revised manuscript in its track changes version.

Editors Comments:

The manuscript needs major revision in order to be accepted for publications in PLOS ONE. Please revise your manuscript considering all points raised by the reviewers.

Response: Thank you for this valuable comment and suggestion, we have carefully revised the manuscript considering all the reviewers’ comments.

Reviewers’ comments:

Reviewers: #1

Let me start by thanking you and the rest of your team for the improved manuscript quality. Overall, you found my comments and questions insightful. I appreciate your positive feedback. I have recommended accepting the revised version of the manuscript. However, it would be best for you and your team to proofread the document for additional typos here and there.

Response: Thank you for your kind consideration and for the valuable comments. We have done our best to improve the manuscript regarding typos and grammar.

Besides, I suggest adding a line or two to account for the study's limitations and list all assumptions made. I still have some reservations that a household head who is usually more on the management and decision-making side would know enough about other tasks completed that are often overlooked by other members who are only sometimes underage (responding to your ethical comment).

Response: We have considered this as a limitation and reflected on it in lines 818-821.

The changes made to the suggested tables and figures and the additional sections for the conclusion and recommendations are all well-suited. The reviewer genuinely appreciates the upcoming extension efforts to raise awareness and prompt behavioral changes through a partnership with the veterinary extension service (VSF-G) and the Ministry of Agriculture, Animal Industry and Fisheries (MAAIF).

Response: Thank you very much.

Reviewer #3

I read the clear version of the revised manuscript appearing first in the pdf, line numbering refers to that version. The introduction introduces the subject, the local setting concerning PPR and RVF, the research questions and the methodologies to be used in a very nice way.

Response: Thank you very much.

Abstract:

Line 30: in the abstract you might want to skip mentioning the name of the software used.

Response: Thank you for the suggestion. This has been removed.

Introduction:

Line 46-49: the sentence is very long, please shorten it or divide it in two.

Response: The sentence has been divided into two, refer to Line 47.

Line 61-62: milking is an important activity that should be mentioned here.

Response: This is an important observation, thank you, and has been added; line 65.

Line 62: I presume the “in” should be placed after “children”.

Response: It has been moved; line 66.

Line 91: instead of “burning and burying” I would prefer “safe disposal of animals that have died from diseases” as there are several ways to do this without spreading disease

Response: This has been revised; Line 104.

Line 93: you need to mention the vaccination guidelines also for PPR.

Response: A national strategy was developed to guide the control and eradication of PPR in Uganda (31) which references different existing policies, such as the Animal Diseases Act, the Cattle Grazing Act and the Cattle Traders Act. Line 105.

Line 93: for RVF, safe handling of abortions (which might present without the animal being “sick)” is also a very important preventive measure, please add this

Response: This has been added. Please refer to line 119.

Line 95-107: this seems not to belong to the subsection PPR and RVF in Uganda but rather to the previous, please move and integrate to make sure to avoid overlap

Response: This has been moved and integrated under the gender and livestock section lines 72-77.

Line 108-132: This also seems not to belong to the subsection PPR and RVF in Uganda but rather to a subsection on gender in decision making and/or decision-making determinant models. Please move/integrate.

Response: This has been moved and integrated into the methods section; refer to lines 134-135

Line 116: which are these five key determinants? Are they only/exactly five? Please clarify.

Response: There are exactly five, including intrapersonal, interpersonal, institutional, and community factors. Please refer to lines 186.

Line 127: what is meant with “all-round” in this context?

Response: This was used to imply that the use of the SEM in livestock disease management allows for an examination of factors at multiple levels and provides a detailed view from individual behaviors to broader policy issues. We have revised this to make it clearer, see lines 140-142 in the methods section.

Line 128: more realistic than what?

Response: More realistic than without including livestock, which was previously rarely done. However, we have revised this in line 140-142.

Line 78: the abbreviation (SR) not previously introduced

Response: This has been corrected at this first mention; please refer to line 97.

Line 87: please remove “as an asset”, or explain why these words are needed here

Response: This has been removed, refer to line 124.

Line 89: please remove “chances of

Response: This has been removed. line 103

Line 133: this seems to be the beginning of another subsection? Please clarify. It would further be a good place to introduce the objectives of the study.

Response: This has been revised as per lines 127-129.

Methodology

Fig 1: I presume that the “corridor region” in the legend refers to the cattle corridor? I further don’t see any reference to the cattle corridor in the text (but I might have missed that), please remove the cattle corridor from the map or mention it in the description of the study area(s).

Response: The cattle corridor region has now been mentioned in the text, please refer to lines 151-154 and reference 49 (Nakiguli et al 2023).

Line 150: please define pastoral/agropastoral

Response: This has been defined. Please refer to lines 166-172 and reference 57 (Otte et al 2020).

Line 153: do you mean disease transmission or disease prevalence here?

Response: We have maintained ‘transmission’ implying that it is high, especially since wild animals are reservoirs to some livestock diseases (lines 173-174).

Line 191-193: the description of the GAHW and CAHW-workforce here seems misplaced, were CAHWs included as key-informants? Please clarify.

Response: The CAHWs were included as key informants since they supported the AHWs within the communities especially in very remote areas that are hard to reach. We have clarified this in lines 243-250.

The description of the methodology, especially data analysis, lacks important clarity and details, see the detailed comments below. Among the missing details not mentioned in the comments below is a section on the researchers’ positionality. As deductive coding is used it is important to reflect and mention the researchers’ previous knowledge and experiences to explain how the analysis process was driven by the researchers’

Response: Thank you for pointing this out. We highlighted the researchers’ positionality in the limitations section at the end of the discussion, please refer to lines 823-829 and reference 68 (Gary A, 2020).

It is further not clear from either the methodology or the results how the SEM and the GAF was used, it now appears that the SEM was only use was to divide the factors influencing disease control decision in the five mentioned categories and there is no description of GAF-analysis in the methodology at all.

Response: The GAF analysis has been added to the methodology section, lines 134-137

Moreover, it is not clear how the deductive codes described in the code books (S1 and S2) was used in the analysis to give the subsections described in the results, or if these subsections are indeed what is described as themes/common themes/analytical themes. This lack in the description of the methodology and the analysis makes it impossible to judge the methodological rigour, and if the conclusions are based on the data.

Line 221-232: please clarify the analysis- process. Did you really code before uploading the data in Nvivo? Did you use deductive coding or were themes emerging from the data? It seems (but this is not clearly described) that deductive coding was used, with the codes used in the respective code book for RVF and PPR), this needs to be described in more details, and in my opinion the codes should be included in the manuscript (not as supplementary material).

Response: Thank you for pointing this out and for the suggestion. We have added additional detail about how these codes were used within the manuscript. Please refer to lines 268-274. The codebooks are still provided in the supplementary materials for readers who prefer the table format.

Line 228: please clarify what you mean by “assess” in this regard (number of, presumed efficiency, quality, cost-benefit etc).

Response: We have reworded to be clearer about the data analysis process; Lines 273-274.

Line 255-258: this belongs in the methods-section. In addition, it is stated here that the factors were organized according to the SEM, which is what I commented on in the general comment, did the SEM include some more analytical elements, or only to organize the factors according to these pre-defined categories?

Response: We apologize for any lack of clarity in the methodology on the raised points. We have tried to be clearer in the methods section now; please refer to lines 289-290.

After reviewing and familiarizing with the transcripts, an initial coding frame was developed, drawing on both the transcript data, the GAF and SEM. The gender activities, resources/services that men and women access to prevent or control disease were identified. The initial codes included: gendered livestock ownership, gendered roles as well as preventive and control measures. The preventive and control measures were sub-coded between men’s and women’s measures (treatment, vector control, culling, movement control and vaccination). All coded data for PPR and RVF data sets were read and re-read to identify connections, similarities or differences between codes. Based on emerging patterns and relationships, codes were organized into overarching themes disaggregated by gender and production system. Unique to RVF were the risky practices/behaviors predisposing animals/humans to zoonoses, but majority of the codes were overlapping PPR and RVF data sets. Thereafter, themes were generated. The themes and the coded data were reviewed to ensure accuracy in the captured data each theme. The measures were then categorized into within farm, between farm and community level depending on where the activity took place. A similar analysis was conducted on the influencing factors, which involved identifying all the factors influencing each choice of a disease preventive and control method. This analysis focused on the underlying factors which determined the gender roles, access to and control over resources/services. These included source of services/information, availability/accessibility/affordability of services, perceptions/views/concerns towards a method, knowledge/awareness, socio-cultural factors, decision-making aspects, trust and experience with a method. The aim of the analysis was to identify and contrast sources of services/resources and to understand their perception/concerns and constraints regarding a particular preventive and control measure. The identified themes were then categorized and analyzed along the social ecological themes by Mcleroy (intrapersonal, interpersonal, institutional, community and policy levels).

I have further have some major concerns about the FGDs, see my comment on S1 below.

Supplementary material 1 (FGD guide): The FGD-guide includes very many and quite detailed questions, even if only considering section A. With section B and C the content becomes enormous. It is written in the S1-file that different FGDs should be used for section A versus sections B and C, but this is not reflected in the manuscript, was those sections used for another study? Please clarify. I also find it unusual to have so many detailed questions in a FGD in which one normally aims at having a free and open discussion driven by the participants. With so many questions the guide looks more like a semi-structured protocol for an individual interview. The length/content of the guide further raises several questions:

Line 205: according to the FGD-guide in S1 the PE-tools were not a complement, but an integrated part of the FGDs? Please clarify and report of the results from the PE-exercises.

How long time did the FGDs take? How was participant engagement ensured during the entire process? How was the results used, as for example the results from the PE are not included in this manuscript?

For question 3a the probing questions and the instruction to the facilitator don’t seem to match (common diseases/all diseases/description of the diseases/zoonotic potential vs ranking based on highest mortality). Were the follow up questions (which are very many) in questions 4-6 asked for all listed diseases? If not, which ones and how was the selection agreed on?

Response: We appreciate the reviewer for all the comments highlighted on the FGD and KII guide. We acknowledge that this was a challenge as each FGD took half a day, but the participants were facilitated with coffee/lunch breaks

We also agree with the comments on the bulkiness of the KII guide, but the questions included questions for all categories of KIIs, e.g. CAHW, AHO, Policy makers etc.

With regards to the other results from the FGDs which are outside the scope of this paper, the information generated through this study was used to write two research briefs one under PPR and another one under RVF. In the attached FGD tool, we have italicized those questions that were used for other outputs

Below are the links to the different project outputs from the FGD findings.

PPR brief: https://hdl.handle.net/10568/118339

RVF brief: https://hdl.handle.net/10568/115766

RVF poster: https://hdl.handle.net/10568/115648

PPR posters: https://hdl.handle.net/10568/115639;
https://hdl.handle.net/10568/124997

Some of the results from the PE especially the one from the resource mapping was used to generate information on resource use (grazing, feeding, marketing and veterinary services) and availability of these resources to both men and women within these communities. Part of this information was used to write a research brief whereas the remaining information was used to build on some of the explanations under the factors influencing disease prevention.

Line 207: you need to add some more information on facilitation, language and translation. Who in the research team facilitated the interviews (FGDs and KIIs)? In what language(s) was the FGDs and KIIs done? By whom in the research team?

Response: Four of the co-authors (Ugandans) facilitated the KIIs in English. Some of the KII were done over the telephone due to COVID restrictions in some of the study regions at the time.

For the FGDs –local facilitators interfaced with the participants and were supervised by one of the Ugandan co-authors. Using the Group Discussion Guide, the trained local facilitators led the FGDs in the local language of each district while the note takers took the notes verbatim. Men facilitated men groups while women facilitated women groups. The FGDs were done in the local languages as described in lines (255-257). We have made changes considering this comment.

By whom in the research team? Was notes taken in English? If not – were the note

---

## [Editor Report · Decision Letter 2]

18 Oct 2024

PONE-D-24-01412R2Gender roles in ruminant disease management in Uganda: Implications for the control of peste des petits ruminants and Rift Valley fever.PLOS ONE

Dear Dr. Jane,

Thank you for submitting your manuscript to PLOS ONE. After careful consideration, we feel that it has merit but does not fully meet PLOS ONE’s publication criteria as it currently stands. Therefore, we invite you to submit a revised version of the manuscript that addresses the points raised during the review process.

**ACADEMIC EDITOR: ** Dear authors, the manuscripts need a major revision for improvement especially the M & M, Results and Discussion sections. Please make the emendments suggested by the reviewers and respond carefully to each point raied by them.

We look forward to receiving your revised manuscript.

Kind regards,

Nussieba A. Osman, Dr. Med. Vet.

Academic Editor

PLOS ONE

---

## [Author Response · Author response to Decision Letter 3]

30 Oct 2024

Dr. Nussieba A. Osman

Academic Editor

PLOS ONE

October 30, 2024

Re: Revision required for manuscript PONE-D-24-01412R2

Dear Sir,

We appreciate your feedback and the opportunity to resubmit a revised manuscript. We also appreciate the reviewers for taking the time to carefully review the manuscript and give detailed and constructive comments. We have incorporated the changes reflecting all the suggestions provided by the editor and reviewers. For ease of reading, we have grouped our responses by section.

Below is our point-by-point response to each comment. If changes were made in the revised manuscript, we indicate line numbers referring to the revised manuscript in its track changes version.

Editors Comments:

The manuscript needs major revision in order to be accepted for publications in PLOS ONE. Please revise your manuscript considering all points raised by the reviewers.

Response: Thank you for this valuable comment and suggestion, we have carefully revised the manuscript considering all the reviewers’ comments.

Reviewers’ comments:

Reviewers: #1

Let me start by thanking you and the rest of your team for the improved manuscript quality. Overall, you found my comments and questions insightful. I appreciate your positive feedback. I have recommended accepting the revised version of the manuscript. However, it would be best for you and your team to proofread the document for additional typos here and there.

Response: Thank you for your kind consideration and for the valuable comments. We have done our best to improve the manuscript regarding typos and grammar.

Besides, I suggest adding a line or two to account for the study's limitations and list all assumptions made. I still have some reservations that a household head who is usually more on the management and decision-making side would know enough about other tasks completed that are often overlooked by other members who are only sometimes underage (responding to your ethical comment).

Response: We have considered this as a limitation and reflected on it in lines 823-826.

The changes made to the suggested tables and figures and the additional sections for the conclusion and recommendations are all well-suited. The reviewer genuinely appreciates the upcoming extension efforts to raise awareness and prompt behavioral changes through a partnership with the veterinary extension service (VSF-G) and the Ministry of Agriculture, Animal Industry and Fisheries (MAAIF).

Response: Thank you very much.

Reviewer #3

I read the clear version of the revised manuscript appearing first in the pdf, line numbering refers to that version. The introduction introduces the subject, the local setting concerning PPR and RVF, the research questions and the methodologies to be used in a very nice way.

Response: Thank you very much.

Abstract:

Line 30: in the abstract you might want to skip mentioning the name of the software used.

Response: Thank you for the suggestion. This has been removed.

Introduction:

Line 46-49: the sentence is very long, please shorten it or divide it in two.

Response: The sentence has been divided into two, refer to Line 47.

Line 61-62: milking is an important activity that should be mentioned here.

Response: This is an important observation, thank you, and has been added; line 65.

Line 62: I presume the “in” should be placed after “children”.

Response: It has been moved; line 66.

Line 91: instead of “burning and burying” I would prefer “safe disposal of animals that have died from diseases” as there are several ways to do this without spreading disease

Response: This has been revised; Line 104.

Line 93: you need to mention the vaccination guidelines also for PPR.

Response: A national strategy was developed to guide the control and eradication of PPR in Uganda (31) which references different existing policies, such as the Animal Diseases Act, the Cattle Grazing Act and the Cattle Traders Act. Line 105.

Line 93: for RVF, safe handling of abortions (which might present without the animal being “sick)” is also a very important preventive measure, please add this

Response: This has been added. Please refer to line 119.

Line 95-107: this seems not to belong to the subsection PPR and RVF in Uganda but rather to the previous, please move and integrate to make sure to avoid overlap

Response: This has been moved and integrated under the gender and livestock section lines 72-77.

Line 108-132: This also seems not to belong to the subsection PPR and RVF in Uganda but rather to a subsection on gender in decision making and/or decision-making determinant models. Please move/integrate.

Response: This has been moved and integrated into the methods section; refer to lines 134-135

Line 116: which are these five key determinants? Are they only/exactly five? Please clarify.

Response: There are exactly five, including intrapersonal, interpersonal, institutional, and community factors. Please refer to lines 186.

Line 127: what is meant with “all-round” in this context?

Response: This was used to imply that the use of the SEM in livestock disease management allows for an examination of factors at multiple levels and provides a detailed view from individual behaviors to broader policy issues. We have revised this to make it clearer, see lines 140-142 in the methods section.

Line 128: more realistic than what?

Response: More realistic than without including livestock, which was previously rarely done. However, we have revised this in line 140-142.

Line 78: the abbreviation (SR) not previously introduced

Response: This has been corrected at this first mention; please refer to line 97.

Line 87: please remove “as an asset”, or explain why these words are needed here

Response: This has been removed, refer to line 124.

Line 89: please remove “chances of

Response: This has been removed. line 103

Line 133: this seems to be the beginning of another subsection? Please clarify. It would further be a good place to introduce the objectives of the study.

Response: This has been revised as per lines 127-129.

Methodology

Fig 1: I presume that the “corridor region” in the legend refers to the cattle corridor? I further don’t see any reference to the cattle corridor in the text (but I might have missed that), please remove the cattle corridor from the map or mention it in the description of the study area(s).

Response: The cattle corridor region has now been mentioned in the text, please refer to lines 151-154 and reference 49 (Nakiguli et al 2023).

Line 150: please define pastoral/agropastoral

Response: This has been defined. Please refer to lines 171174 and reference 57 (Otte et al 2020).

Line 153: do you mean disease transmission or disease prevalence here?

Response: We have maintained ‘transmission’ implying that it is high, especially since wild animals are reservoirs to some livestock diseases (lines 175-176).

Line 191-193: the description of the GAHW and CAHW-workforce here seems misplaced, were CAHWs included as key-informants? Please clarify.

Response: The CAHWs were included as key informants since they supported the AHWs within the communities, especially in very remote areas that are hard to reach. We have clarified this in lines 249-251.

The description of the methodology, especially data analysis, lacks important clarity and details, see the detailed comments below. Among the missing details not mentioned in the comments below is a section on the researchers’ positionality. As deductive coding is used it is important to reflect and mention the researchers’ previous knowledge and experiences to explain how the analysis process was driven by the researchers’

Response: Thank you for pointing this out. We highlighted the researchers’ positionality in the limitations section at the end of the discussion, please refer to lines 826-832 and reference 68 (Gary A, 2020).

It is further not clear from either the methodology or the results how the SEM and the GAF was used, it now appears that the SEM was only use was to divide the factors influencing disease control decision in the five mentioned categories and there is no description of GAF-analysis in the methodology at all.

Response: The GAF analysis has been added to the methodology section, lines 134-137

Moreover, it is not clear how the deductive codes described in the code books (S1 and S2) was used in the analysis to give the subsections described in the results, or if these subsections are indeed what is described as themes/common themes/analytical themes. This lack in the description of the methodology and the analysis makes it impossible to judge the methodological rigour, and if the conclusions are based on the data.

Line 221-232: please clarify the analysis- process. Did you really code before uploading the data in Nvivo? Did you use deductive coding or were themes emerging from the data? It seems (but this is not clearly described) that deductive coding was used, with the codes used in the respective code book for RVF and PPR), this needs to be described in more details, and in my opinion the codes should be included in the manuscript (not as supplementary material).

Response: Thank you for pointing this out and for the suggestion. We have added additional detail about how these codes were used within the manuscript. Please refer to lines 270-279. The codebooks are still provided in supplementary materials for readers who prefer the table format.

Line 228: please clarify what you mean by “assess” in this regard (number of, presumed efficiency, quality, cost-benefit etc).

Response: We have reworded to be clearer about the data analysis process; Lines 279-279.

Line 255-258: this belongs in the methods section. In addition, it is stated here that the factors were organized according to the SEM, which is what I commented on in the general comment, did the SEM include some more analytical elements, or only organize the factors according to these pre-defined categories?

Response: We apologize for any lack of clarity in the methodology on the points raised. We have tried to be clearer in the methods section now; please refer to lines 291-292.

After reviewing and familiarizing with the transcripts, an initial coding frame was developed, drawing on both the transcript data, the GAF and SEM. The gender activities, resources/services that men and women access to prevent or control disease were identified. The initial codes included: gendered livestock ownership, gendered roles as well as preventive and control measures. The preventive and control measures were sub-coded between men’s and women’s measures (treatment, vector control, culling, movement control and vaccination). All coded data for PPR and RVF data sets were read and re-read to identify connections, similarities or differences between codes. Based on emerging patterns and relationships, codes were organized into overarching themes disaggregated by gender and production system. Unique to RVF were the risky practices/behaviors predisposing animals/humans to zoonoses, but majority of the codes were overlapping PPR and RVF data sets. Thereafter, themes were generated. The themes and the coded data were reviewed to ensure accuracy in the captured data each theme. The measures were then categorized into within farm, between farm and community level depending on where the activity took place. A similar analysis was conducted on the influencing factors, which involved identifying all the factors influencing each choice of a disease preventive and control method. This analysis focused on the underlying factors which determined the gender roles, access to and control over resources/services. These included source of services/information, availability/accessibility/affordability of services, perceptions/views/concerns towards a method, knowledge/awareness, socio-cultural factors, decision-making aspects, trust and experience with a method. The aim of the analysis was to identify and contrast sources of services/resources and to understand their perception/concerns and constraints regarding a particular preventive and control measure. The identified themes were then categorized and analyzed along the social ecological themes by Mcleroy (intrapersonal, interpersonal, institutional, community and policy levels).

I have further have some major concerns about the FGDs, see my comment on S1 below.

Supplementary material 1 (FGD guide): The FGD-guide includes very many and quite detailed questions, even if only considering section A. With section B and C the content becomes enormous. It is written in the S1-file that different FGDs should be used for section A versus sections B and C, but this is not reflected in the manuscript, was those sections used for another study? Please clarify. I also find it unusual to have so many detailed questions in a FGD in which one normally aims at having a free and open discussion driven by the participants. With so many questions the guide looks more like a semi-structured protocol for an individual interview. The length/content of the guide further raises several questions:

Line 205: according to the FGD-guide in S1 the PE-tools were not a complement, but an integrated part of the FGDs? Please clarify and report of the results from the PE-exercises.

How long did the FGDs take? How was participant engagement ensured during the entire process? How was the results used, for example the results from the PE are not included in this manuscript?

For question 3a the probing questions and the instruction to the facilitator don’t seem to match (common diseases/all diseases/description of the diseases/zoonotic potential vs ranking based on highest mortality). Were the follow up questions (which are very many) in questions 4-6 asked for all listed diseases? If not, which ones and how was the selection agreed on?

Response: We appreciate the reviewer for all the comments highlighted on the FGD and KII guide. We acknowledge that this was a challenge as each FGD took half a day, but the participants were facilitated with coffee/lunch breaks

We also agree with the comments on the bulkiness of the KII guide, but the questions included questions for all categories of KIIs, e.g. CAHW, AHO, Policy makers etc.

With regards to the other results from the FGDs which are outside the scope of this paper, the information generated through this study was used to write two research briefs, one under PPR and another one under RVF. In the attached FGD tool, we have italicized those questions that were used for other outputs

Below are the links to the different project outputs from the FGD findings.

PPR brief: https://hdl.handle.net/10568/118339

RVF brief: https://hdl.handle.net/10568/115766

RVF poster: https://hdl.handle.net/10568/115648

PPR posters: https://hdl.handle.net/10568/115639;
https://hdl.handle.net/10568/124997

Some of the results from the PE especially the one from the resource mapping was used to generate information on resource use (grazing, feeding, marketing and veterinary services) and availability of these resources to both men and women within these communities. Part of this information was used to write a research brief whereas the remaining information was used to build on some of the explanations under the factors influencing disease prevention.

Line 207: you need to add some more information on facilitation, language and translation. Who in the research team facilitated the interviews (FGDs and KIIs)? In what language(s) was the FGDs and KIIs done? By whom in the research team?

Response: Four of the co-authors (Ugandans) facilitated the KIIs in English. Some of the KII were done over the telephone due to COVID restrictions in some of the study regions at the time.

For the FGDs –local facilitators interfaced with the participants and were supervised by one of the Ugandan co-authors. Using the Group Discussion Guide, the trained local facilitators led the FGDs in the local language of each district while the note takers took the notes verbatim. Men facilitated men groups while women facilitated women groups. The FGDs were done in the local languages as described in lines (257-259). We have made changes considering this comment.

By whom in the research team? Was notes taken in English? If not – were the notes translated?

---

## [Decision Letter · Decision Letter 3]

6 Nov 2024

PONE-D-24-01412R3Gender roles in ruminant disease management in Uganda: Implications for the control of peste des petits ruminants and Rift Valley fever.PLOS ONE

Dear Dr. Jane,

Thank you for submitting your manuscript to PLOS ONE. After careful consideration, we feel that it has merit but does not fully meet PLOS ONE’s publication criteria as it currently stands. Therefore, we invite you to submit a revised version of the manuscript that addresses the points raised during the review process.

**ACADEMIC EDITOR:** Dear authors, Many comments raised by the reviewer were answered in the response to the reviewer's comments but need to be incorporated in the text. Please revise the manuscript carefully, trying to incorporate all these answers in addition to the new comments raised by the reviwer.

We look forward to receiving your revised manuscript.

Kind regards,

Nussieba A. Osman, Dr. Med. Vet.

Academic Editor

PLOS ONE

Journal Requirements:

Reviewers' comments:

Reviewer's Responses to Questions

**Comments to the Author**

1. If the authors have adequately addressed your comments raised in a previous round of review and you feel that this manuscript is now acceptable for publication, you may indicate that here to bypass the “Comments to the Author” section, enter your conflict of interest statement in the “Confidential to Editor” section, and submit your "Accept" recommendation.

Reviewer #3: (No Response)

2. Is the manuscript technically sound, and do the data support the conclusions?

Reviewer #3: Yes

3. Has the statistical analysis been performed appropriately and rigorously? 

Reviewer #3: N/A

4. Have the authors made all data underlying the findings in their manuscript fully available?

Reviewer #3: Yes

5. Is the manuscript presented in an intelligible fashion and written in standard English?

Reviewer #3: Yes

6. Review Comments to the Author

Reviewer #3: Many of my questions and comments have been answered but not all. Some has been answered in the point-to-point-answers to me, but with no changes done to the text. Generally, if a question is posed in the reviewer report, it implies tat changes are needed in the text, it will seldom be sufficient to just provide an answer to the reviewer.

Questions from previous review that need to be answered or incorporated in the text and not only answered in the point-to-point answers:

How/by whom in the research team translated the interview guides?

Who in the research team facilitated the interviews (FGDs and KIIs)?

In what language(s) was the FGDs and KIIs done?

How long time did the FGDs take?

How was participant engagement ensured during the entire process?

How was the results used, as for example the results from the PE are not included in this manuscript?

For question 3a the probing questions and the instruction to the facilitator don’t seem to match (common diseases/all diseases/description of the diseases/zoonotic potential vs ranking based on highest mortality). Were the follow up questions (which are very many) in questions 4-6 asked for all listed diseases?

If not, for which ones and how was the selection agreed on?

According to the FGD-guide in S1 the PE-tools were not a complement, but an integrated part of the FGDs? Please clarify and report of the results from the PE-exercises.

New questions/comments:

Line 32: there seems to be an abundant “the” in this sentence.

Line 48: the sentence is long with several sub-clauses and contains two “especially”, please rephrase.

Line 59: isn’t “vaccination and breeding” part of animal husbandry? Maybe you mean “daily care” or something similar?

Line 61: this sentence seems to discuss the same theme as the previous so I think the blank line should be omitted.

Line 77-81: please check this sentence for grammar/synthax.

Line 82: this sentence seems to discuss the same theme as the previous so I think the blank line should be omitted.

Line 89: does this statement only refer to one RVF-vaccine (“the RVF vaccine” (and which in that case), or should it be ”RVF-vaccines”?

Line 105-107: please check this sentence for grammar/synthax.

Line 124: ”as an investment or savings” seems redundant here, please remove.

Line 136 and 138: please use the abbreviations (GAF and SEM) all through once introduced.

Line 136: do you mean “elders” or “the elderly”? Please clarify.

Line 209: a “selected” seems to be missing.

Line 260: were the facilitators part of the research team?

Line 291: Is the “socio-ecological themes” the same as the SEM?

Line 291: Please remove the redundant full stop.

Line 290 and 202: are “identified themes” and “selected themed” the same?

Lines 316-319: it seems like the GAF (or part thereof, see below) was applied to the first section and the SEM to the second section? Was this the case? If so, please clarify this in the M&M and results.

Lines 322-325: are these results? If so please check the tempus used, if not, move to introduction.

Lines 329-331: I don’t understand what you mean with this sentence, isn’t that what you will do using the GAF and SEM?

Line 345-387: this section seems to analyse/describe the different measures and tasks performed by men and women and to some extent boys and girls. If this is meant to represent the GAF-analysis, you need to mention somewhere that you analyse according to GAF – minus the elderly.

Lines 380-382: see my comment on this in the previous report

Line 475: Please change to “foot and mouth disease”.

Line 680: I think “vaccine” should be “vaccinate”?

Line 703: There seems to be a “what” missing after “asked”.

Line 705-708: Isn’t that what you described in your results? Here in the discussion you are meant to discuss these results against the literature.

Line 717: suggestion: “might be limited for all categories (men, women, boys and girls) of participants”. Or did you see any gender differences in this regard?

Line 782: And themselves!

Lines 782-784: Maybe worth mentioning costs of transport and women’s restricted access to monetary resources here if you think it is relevant based on your results.

Lines 789-795: Maybe worth mentioning here that women faced specific challenge in accessing vaccination, not only because of being smallholders, but specifically related to gender? That is at least how I interpreted you results on this?

Line 832 and forward: this section is length and mostly contains policy recommendations. For clarity, I would recommend dividing it into (scientific) conclusions of the study, and policy recommendations regarding vaccination and disease control in Uganda (and similar contexts).

Table 1: Please check the formatting.

7. PLOS authors have the option to publish the peer review history of their article (what does this mean? ). If published, this will include your full peer review and any attached files.

**Do you want your identity to be public for this peer review?** For information about this choice, including consent withdrawal, please see our Privacy Policy .

Reviewer #3: No

---

## [Author Response · Author response to Decision Letter 4]

25 Dec 2024

Dr. Nussieba A. Osman

Academic Editor

PLOS ONE

December 25, 2024

Re: Revision required for manuscript PONE-D-24-01412R3

Dear Dr. Osman,

We would like to thank the academic editor and the reviewer once again for taking the time to review our manuscript and for the relevant remarks and comments which have helped us to improve the quality of our paper. We have done a careful re-reading of our paper and addressed the comments raised. We have also corrected some typographical errors.

These changes are detailed below.

Editor’s Comments:

Please review your reference list to ensure that it is complete and correct. If you have cited papers that have been retracted, please include the rationale for doing so in the manuscript text or remove these references and replace them with relevant current references.

Any changes to the reference list should be mentioned in the rebuttal letter that accompanies your revised manuscript. If you need to cite a retracted article, indicate the article’s retracted status in the References list and also include a citation and full reference for the retraction notice.

Response: Thanks for these valuable comments. We have rechecked the reference list thoroughly and found one reference (ref 10) Dudi et al (2019) which we have removed as it is no longer available online.

Reviewer’s comments:

Introduction

Line 59: isn’t “vaccination and breeding” part of animal husbandry? Maybe you mean “daily care” or something similar?

Response: Thank you for highlighting this. We have revised manuscript; The roles of women have been broken down to feeding animals, fodder collection, and cleaning animal sheds. Lines 58-60.

Line 89: does this statement only refer to one RVF-vaccine (“the RVF vaccine” (and which in that case), or should it be” RVF-vaccines”?

Response: This has been revised to read “vaccines” as suggested. Line 92.

Line 124:as an investment or savings” seems redundant here, please remove.

Response: We have maintained investment: Please refer to Line 128.

Methodology

Many of my questions and comments have been answered but not all. Some has been answered in the point-to-point-answers to me, but with no changes done to the text.

Generally, if a question is posed in the reviewer report, it implies that changes are needed in the text, it will seldom be sufficient to just provide an answer to the reviewer.

Questions from previous review that need to be answered or incorporated in the text and not only answered in the point-to-point answers:

Response: We appreciate your specifications of the pending questions posed in the previous review and not indicated in the text. We have made changes considering the comments highlighted.

How/by whom in the research team translated the interview guides? In what language(s) was the FGDs and KIIs done? How long time did the FGDs take?

Response: We have incorporated the changes below in the new version of the manuscript.

Lines 226-231: Prior to conducting the research activities, a team of four people fluent in the respective local languages (two elders, two animal health practitioners from each study district) translated the study guide. The translated guide was further refined by the local facilitators from each district two-day training workshop. The enumerators were taken through the questions and associated probes to ensure that they understood the purpose and intent of each question.

Regarding who facilitated the interviews (FGDs and KIIs) in the research team:

The trained local facilitators led the FGDs in the local language of each district while the note takers took the notes verbatim. Men facilitated men’s groups while women facilitated women’s groups. The local facilitators and note takers were supervised by one of the Ugandan co-authors. Using the FGD guide, four of the co-authors (Ugandans) facilitated the KIIs in English. Some of the KII were done over the telephone due to COVID restrictions in some of the study regions at the time.

For the FGDs, local facilitators interfaced with the participants and were supervised by one of the Ugandan co-authors. Using the Group Discussion Guide, the trained local facilitators led the FGDs in the local language of each district while the note takers took the notes verbatim.

The FGDs were done in the local languages as described in lines (281-283):

Karamojong for Napak and Nakapiripirit, Runyankole for Isingiro, Luganda for Sembabule, Ateso for Serere, and Luganda, Ateso, or Lugweere for Butebo.

According to the FGD-guide in S1 the PE-tools were not a complement, but an integrated part of the FGDs? Please clarify and report of the results from the PE-exercises.

Response: We agree with the reviewer’s comment. The PE tools were part of the FGDs. Some of the results from the PE, especially resource mapping, were used to generate information on resource use (grazing, feeding, marketing and veterinary services) and availability of these resources to both men and women within these communities. Since the FGD tool covered several aspects and served several purposes, part of this information was used to build on some of the explanations under the factors influencing disease prevention. The PE results were also used to write research briefs and non-technical policy briefs for the Ministry of Agriculture, Animal Industries and Fisheries in Uganda.

How was the results used, for example the results from the PE are not included in this manuscript?

Response: With regards to the other results from the FGDs which are outside the scope of this paper, the information generated through this study was used to write two research briefs, one under PPR and another one under RVF, posters and blogs as indicated below; Please refer to lines 258-259.

Reference number Reference

62 Asindu et al (2021)

63 Asindu et al (2021)

64 Lule et al (2022)

65 Lule et al (2021)

66 Namatovu et al (2022)

In the attached FGD tool, we have italicized those questions that were used for other outputs.

How was participant engagement ensured during the entire process?

Response: Various participatory tools were used to facilitate discussions and ensure participant engagement. These included simple ranking and proportional piling. Please refer to lines 241-244 (references 60 and 61): Energizers and coffee/lunch breaks were provided in between sessions.

For question 3a the probing questions and the instruction to the facilitator don’t seem to match (common diseases/all diseases/description of the diseases/zoonotic potential vs ranking based on highest mortality). Were the follow up questions (which are very many) in questions 4-6 asked for all listed diseases? If not, which ones and how was the selection agreed on?

Response: We appreciate your constructive critique, which helps us to clarify more. This was an error that was corrected. Refer to S1 File

As regards whether the follow-up questions were asked, these were asked with the intent of exploring awareness of livestock keepers on livestock diseases affecting their SR/Cattle including PPR and RVF. Please refer to lines 235-240 and 253-254.

First, participants listed the common disease problems that affected their goats/sheep/cattle and described the signs of each disease (local disease names were noted). For the ranking of diseases, 100 beans/counters were distributed among the diseases to indicate their relative importance. In the process, participants were asked to explain the reasons for assigning the scores. The participants were asked to identify the causes of the diseases mentioned, which diseases caused more deaths, and which age groups were most affected. Participants were then asked to explain how each disease was spreading between animals and whether it could be transmitted between animals and people, and who of the household was involved in activities that might lead to disease transmission. Ways of transmission were listed (within farm, between farm, community level). Participants were asked to explain what they did in response to the mentioned diseases to understand their disease coping strategies

Line 136 and 138: please use the abbreviations (GAF and SEM) all through once introduced.

Response: This has been revised, the acronyms are introduced in lines 138-140 then used throughout the manuscript.

Line 136: do you mean “elders” or “the elderly”? Please clarify.

Response: This has been revised to elderly. Please refer to line 141.

Line 260: were the facilitators part of the research team?

Yes, they were part of the research team. Please refer to line 287-288.

Line 291: Is the “socio-ecological themes” the same as the SEM?

Response: Yes, this has been revised to SEM.

Line 290 and 202: are “identified themes” and “selected themed” the same?

Response: This has been revised to selected theme to imply that predetermined themes were applied to the factors influencing livestock disease prevention and control. Please refer to line 317

Lines 316-319: it seems like the GAF (or part thereof, see below) was applied to the first section and the SEM to the second section? Was this the case? If so, please clarify this in the M&M and results.

Response: This has been clarified in the Methods section, lines 293-294.

Lines 322-325: are these results? If so please check the tempus used, if not, move to introduction.

Response: Yes, these are results. We have revised the sentence to make it clearer. (lines 347-351).

Lines 329-331: I don’t understand what you mean with this sentence, isn’t that what you will do using the GAF and SEM?

Response: Yes, this is what is done with the GAF and SEM. The sentence has been deleted.

Line 345-387: this section seems to analyse/describe the different measures and tasks performed by men and women and to some extent boys and girls. If this is meant to represent the GAF-analysis, you need to mention somewhere that you analyse according to GAF – minus the elderly.

Response: Thank you for the suggestion. This has been added in line 342-343.

Lines 380-382: see my comment on this in the previous report:

Comment from the previous review: I confess to not having read the article in reference, but disregarding what it says, the association between purchasing acaricides and domestic violence and in the next step suicide needs to be more thoroughly introduced here:

Response: We have thoroughly explained the connection between acaricide use, domestic violence, and suicidal tendences among women in Karamoja as follows.

Lines 88-91: Given the high incidences of domestic violence against women in Karamoja (22), and cases of women using acaricides to poison themselves to escape from domestic abuse, women have been banned from purchasing acaricides (20).

Lines 403-405: Some veterinary drug shops in the pastoral areas of Karamoja have banned the selling of acaricides and associated chemicals to women due to reports of suicidal thoughts among women in these areas as a result of domestic abuse and marital conflicts (20).

Line 475: Please change to “foot and mouth disease”.

Response: Revised accordingly. Line 499

Line 680: I think “vaccine” should be “vaccinate”?

Response: Thanks very much for the suggestion. This has been revised to vaccinate.

Line 705-708: Isn’t that what you described in your results? Here in the discussion you are meant to discuss these results against literature.

Response: Thanks very much for highlighting this. The whole sentence has been deleted.

Line 717: suggestion: “might be limited for all categories (men, women, boys and girls) of participants”. Or did you see any gender differences in this regard?

Response: The gender specific difference has been captured within the text. Please refer to line 737, reference 66, Namatovu et al (2022): reference 69, Namatovu et al (2021)

Line 782: And themselves!

Response: Thank you for the suggestion. This has been added.

Lines 782-784: Maybe worth mentioning costs of transport and women’s restricted access to monetary resources here if you think it is relevant based on your results.

Response: Thanks very much for this valuable suggestion. We have expanded on the explanation within the text and taken into consideration women’s restricted monetary resources which might prevent them from meeting related transport costs (Lines 803-806

Lines 789-795: Maybe worth mentioning here that women faced specific challenge in accessing vaccination, not only because of being smallholders, but specifically related to gender? That is at least how I interpreted you results on this?

Response: As per the suggestion, we have mentioned the gender specific challenges within the text. Please refer to lines 802-809

Line 832 and forward: this section is length and mostly contains policy recommendations.

For clarity, I would recommend dividing it into (scientific) conclusions of the study, and policy recommendations regarding vaccination and disease control in Uganda (and similar contexts).

Response: We highly appreciate your guidance on this. We have categorized the recommendations and conclusions into policy and scientific conclusions.

Table 1: Please check the formatting.

Response: This has been formatted

Typographical errors

We have corrected the typographical errors and made some improvements to obtain a more readable version as indicated below.

Line 32: there seems to be an abundant “the” in this sentence.

Response: This has been deleted.

Line 77-81: please check this sentence for grammar/syntax.

Response: the whole sentence has been revised.

Line 48: the sentence is long with several sub-clauses and contains two “especially”, please rephrase.

Response: The sentence has been revised.

Line 82: this sentence seems to discuss the same theme as the previous so I think the blank line should be omitted.

Response: Thank you. This has been revised

Line 61: this sentence seems to discuss the same theme as the previous so I think the blank line should be omitted.

Thanks very much for these valuable suggestions, the blank line has been deleted.

Line 209: a “selected” seems to be missing.

Response: Thank you for highlighting this. The word ‘selected’ has been added. Please refer to line 212

Line 291: Please remove the redundant full stop.

Response: This has been deleted.

Line 105-107: please check this sentence for grammar/synthax.

Response: The sentence has been revised. Please refer to lines 109-111

Line 703: There seems to be a “what” missing after “asked”.

Response: Thank you, this has been added and the whole sentence revised

Sincerely,

Jane Namatovu, Peter Lule, Marsy Asindu, Zoë A. Campbell, Dan Tumusiime, Henry Kiara, Bernard Bett, Kristina Roesel, Emily Ouma

---

## [Decision Letter · Decision Letter 4]

6 Jan 2025

PONE-D-24-01412R4Gender roles in ruminant disease management in Uganda: Implications for the control of peste des petits ruminants and Rift Valley fever.PLOS ONE

Dear Dr. Jane,

Thank you for submitting your manuscript to PLOS ONE. After careful consideration, we feel that it has merit but does not fully meet PLOS ONE’s publication criteria as it currently stands. Therefore, we invite you to submit a revised version of the manuscript that addresses the points raised during the review process. Please submit your revised manuscript by Feb 20 2025 11:59PM. If you will need more time than this to complete your revisions, please reply to this message or contact the journal office at plosone@plos.org . Please include the following items when submitting your revised manuscript:

We look forward to receiving your revised manuscript.

Kind regards,

Nussieba A. Osman, Dr. Med. Vet.

Academic Editor

PLOS ONE

Journal Requirements:

**Additional Editor Comments:**

Dear authors, please carefully make the amendment suggested by the third reviewer. After that your manuscript will be considered accepted for publications in PLOS One.

Reviewers' comments:

Reviewer's Responses to Questions

**Comments to the Author**

1. If the authors have adequately addressed your comments raised in a previous round of review and you feel that this manuscript is now acceptable for publication, you may indicate that here to bypass the “Comments to the Author” section, enter your conflict of interest statement in the “Confidential to Editor” section, and submit your "Accept" recommendation.

Reviewer #3: (No Response)

2. Is the manuscript technically sound, and do the data support the conclusions?

Reviewer #3: Yes

3. Has the statistical analysis been performed appropriately and rigorously? 

Reviewer #3: N/A

4. Have the authors made all data underlying the findings in their manuscript fully available?

Reviewer #3: Yes

5. Is the manuscript presented in an intelligible fashion and written in standard English?

Reviewer #3: Yes

6. Review Comments to the Author

Reviewer #3: General comments

The manuscript has improved a lot since the last revision, some minor, mainly editorial comments remain/have arisen due to changes in the text, see details with line numbers below.

Apart from the minor comments I have a major concern with the discussion. It still lacks engagement with the literature, and a large part of it is policy, veterinary extension and scientific recommendations. The latter is obviously an important output from research, but as this is a scientific contribution the equilibrium between recommendations and scientific discussion needs to better balanced. As the study has already produced other outputs (including recommendations), maybe the recommendations can be reduced here? Or included in another form (as a scientific discussion). As an example of the lack of engagement with the literature large parts of the discussion discuss the results, but without referring to the literature, e.g lines 748-774, 778-798, 811-820, 823-830.

Minor comments

Line 5: A comma seems to be missing between the numbers for the affiliation 1 and 5.

Line 59: Please replace the comma after “fodder collection” with “and”.

Line 219-225: Please state in which district the “extra” 4 FGD was held, and why (6 districts*2 subcounties in each*2FGD in each soubcounty (one men and one women)=24, but it seems like 28 FGDs were held?).

Line 229: The sentence is not complete, please revise.

Line 232-233: It seems like the sentence starting on line 232 belongs to the previous sub-section, and that the sentence starting on line 233 belongs to a new subsection, i.e. that a blank line should be introduced between these two sentences? Please revise.

Line 250: What do you mean with “various aspects of the FGD-guide” in this context? Do you mean “since the FGD-guide covered many topics” or something similar? Please revise.

Line 252: Please replace the comma after “resource mapping” with “and”.

Line 259: Please remove “BUILD” from the sentence.

Line 339/Table 1: Please write out in the caption or as a foot note to the table what you mean by # (or write out “number of” in full).

Line 342-343: Please revise the sentence, the “minus the elderly” reads somewhat bizarre.

Line 347-349: Please revise the sentence.

Line 371: Please remove the comma after “isolation of sick animals”.

Line 381: Please remove the comma after “sick”.

Line 407 and other places: Please check how you mark the start of a new subsection (new line, indent, blank line) and use the same all through the manuscript.

Line 413-415: Please use italics for all citations, and use indent (or not) for all citations.

Line 458: Please remove the comma after “vendors”.

Line 469: This is confusing, restricting access to neighbours’ animals/movement seems to imply a belief that disease are spread by direct contact and not by ticks? Pease clarify.

Line 473-474: Do you mean in both areas? Pease clarify.

Line 497: Please write out numbers less than twelve in letters.

Line 499/502: Is the different use of the abbreviation/disease name in full in these two quotes a mirror of how the discussants used terms, or an oversight from the authors? Please check this up.

Line 524: Please remove the comma after “disease restrictions”.

Line 528: Please remove the comma after “lameness and death”.

Line 530: Please revise the sentence (remove the subclausation).

Line 619: Please place the reference at the end of the sentence.

Line 727: Maybe “investigated” or similar would be a more fitting verb than “asked”? Please consider rewording.

Line 762: Please use the abbreviation (SR) once introduced.

Line 818-820: Please revise the sentence, now it is written to imply that women themselves are affected by PPR.

Line 858-861: This seems to be more of a summary than a conclusion. Please move to a more relevant section.

7. PLOS authors have the option to publish the peer review history of their article (what does this mean? ). If published, this will include your full peer review and any attached files.

**Do you want your identity to be public for this peer review?** For information about this choice, including consent withdrawal, please see our Privacy Policy .

Reviewer #3: No

---

## [Author Response · Author response to Decision Letter 5]

27 Jan 2025

Dr. Nussieba A. Osman

Academic Editor

PLOS ONE

January 27, 2025

RE: Revision for Manuscript PONE-D-24-01412R4

Dear Dr. Osman,

Thank you for giving us an opportunity to revise our manuscript for publication. We appreciate the time you and the reviewer have spent contributing feedback for us to strengthen our paper. We have made the following amendments to reflect the comments and suggestions.

Editor Comments:

Comment: Please review your reference list to ensure that it is complete and correct. If you have cited papers that have been retracted, please include the rationale for doing so in the manuscript text or remove these references and replace them with relevant current references. Any changes to the reference list should be mentioned in the rebuttal letter that accompanies your revised manuscript. If you need to cite a retracted article, indicate the article’s retracted status in the References list and also include a citation and full reference for the retraction notice.

Response: Thank you for the comment. We have checked the reference list and updated the accession numbers for the following references.

Reference Reference number

Flintan, F (2008) 15

The Cross Cultural Foundations of Uganda (2020) 22

Nkamwesiga, J. et al (2022) 26

WOAH (2018) 35

Baudin, M. et al (2016) 37

Doran, K. et al (2017) 45

Sadarangani, TR. (2020) 47

March, C. et al (1999) 48

Nkamwesiga, J. et al (2020) 52

Nkamwesiga, J. et al. (2019) 53

Tumusiime, D. et al (2019) 55

Alemu et al (2019) 60

BOU (2021) 68

Dumas, SE (2018) 69

Reviewer #3 comments:

Comment: The manuscript has improved a lot since the last revision, some minor, mainly editorial comments remain/have arisen due to changes in the text, see details with line numbers below.

Response: Thank you very much.

Comment: Apart from the minor comments I have a major concern with the discussion. It still lacks engagement with literature, and a large part of it is policy, veterinary extension and scientific recommendations. The latter is obviously an important output from research, but as this is a scientific contribution the equilibrium between recommendations and scientific discussion needs to better balanced. As the study has already produced other outputs (including recommendations), maybe the recommendations can be reduced here? Or included in another form (as a scientific discussion). As an example of the lack of engagement with the literature large parts of the discussion discuss the results, but without referring to the literature, e.g lines 748-774, 778-798, 811-820, 823-830.

Response: We appreciate your direction. We have revised the discussion and tried to engage more with existing gender and veterinary literature by incorporating the following references.

Reference Reference number Line number

United States Department of Agriculture (2012) 75 751

Dione, M. (2021) 76 757

Bishop, RP. et al (2023) 77 760

Nuvey, FS. et al (2022) 78 762

Serra, R. et al (2022) 79 765

Shibata, R. et al (2020) 80 765

Mburu, CM. et al (2023) 81 777

Robi, DT. et al (2023) 82 779

Williams, S. et al (2022) 83 780

Ahmed, H. et al (2021) 84 782

Bjornlund, H. et al (2019) 86 791

Krause, BL (2024) 87 793

Malapit, H. et al (2020) 88 793

Po, JYT et al (2020) 89 793

Mutua, E. et al (2017) 90 806

Kiara, H. et al (2017) 93 833

Conrad, B. et al (2024) 94 835

IDRC (2023) 96 888

Comment: Line 219-225: Please state in which district the “extra” 4 FGD was held, and why (6 districts*2 subcounties in each*2FGD in each soubcounty (one men and one women) =24, but it seems like 28 FGDs were held?).

Response: Thank you so much for highlighting this. In short, the PPR sites covered three districts and the RVF sites covered four districts (lines 163-165) , which led to 12 and 16 FGDs respectively.

We added some text to clarify:

Lines 210-214

“In total, 28 FGDs were conducted: 16 FGDs with women and men keeping cattle in the RVF districts, and 12 FGDs with women and men keeping small ruminants in the PPR sites. The numbers were slightly different because the RVF sites had four enrolled districts and the PPR sites had only three, with Isingiro District selected for both diseases.”

Comment: Line 232-233: It seems like the sentence starting on line 232 belongs to the previous sub-section, and that the sentence starting on line 233 belongs to a new subsection, i.e. that a blank line should be introduced between these two sentences? Please revise.

Response: This has been revised. Please refer to lines 234 and 237.

Comment: Line 250: What do you mean with “various aspects of the FGD-guide” in this context? Do you mean “since the FGD-guide covered many topics” or something similar? Please revise.

Response: Thank you for pointing this out. This has been revised to “Since the FGD guide covered many topics”, Line 254

Comment: Line 469: This is confusing, restricting access to neighbours’ animals/movement seems to imply a belief that disease are spread by direct contact and not by ticks? Pease clarify.

Response: We apologize for the lack of clarity. We have revised the sentence to make it clearer: Line 472

“Men in the mixed and agropastoral systems believed that direct contact led to spread of tick-borne diseases to their livestock, and as a result, took action to restrict animal movement to avoid contact with livestock from neighboring farms.”

Comment: Line 473-474: Do you mean in both areas? Pease clarify.

Response: This has been revised to indicate the difference in practices in the two areas. Line 477-478

“Fencing individually owned farms at household level in agropastoral and relocating animal kraals in pastoral areas were done more for cattle than for SRs.”

Comment: Line 499/502: Is the different use of the abbreviation/disease name in full in these two quotes a mirror of how the discussants used terms, or an oversight from the authors? Please check this up.

Response: Thanks for pointing this out. This has been revised within the text to mirror the term used by discussants in the two quotes. The discussants referred to the disease name as ‘FMD’. This level of familiarity with the disease due to the vaccination campaigns seemed worth mentioning, so we also added an additional line of text about it. Lines 503-504

“The respondents were familiar enough with FMD to refer to it directly using the abbreviation as shown in the quotes below.”

Comment: Line 727: Maybe “investigated” or similar would be a more fitting verb than “asked”? Please consider rewording.

Response: This has been revised. Line 734

“This study investigated what men and women did to prevent or control diseases that affected them and their livestock as well as the factors that influenced the choice of disease control measures.”

Comment: Line 818-820: Please revise the sentence, now it is written to imply that women themselves are affected by PPR.

Response: This has been revised: Lines 849

“Given the low awareness of RVF and PPR, any recommendations about prevention and control were inferred from how people prevented or controlled other ruminant diseases.”

Comment: Line 858-861: This seems to be more of a summary than a conclusion. Please move to a more relevant section.

Response: This has been moved to the discussion section. Lines 836-840

Comment: Line 342-343: Please revise the sentence, the “minus the elderly” reads somewhat bizarre.

Response: Agreed, this has been deleted. Line 345

“We present the results of the thematic analysis in two sections: The first section uses GAF, and the second section uses SEM.”

Also added as a limitation from line 860.

“GAF structure includes the elderly for consideration as a separate category but there were not enough elderly respondents for separate analysis.”

Comment: Line 347-349: Please revise the sentence.

Response: This has been revised. Line 349

Typological errors

Line 5: A comma seems to be missing between the numbers for the affiliation 1 and 5.

Line 59: Please replace the comma after “fodder collection” with “and”.

Line 229: The sentence is not complete, please revise.

Line 252: Please replace the comma after “resource mapping” with “and”.

Line 259: Please remove “BUILD” from the sentence.

Line 339/Table 1: Please write out in the caption or as a foot note to the table what you mean by # (or write out “number of” in full).

Line 371: Please remove the comma after “isolation of sick animals”.

Line 381: Please remove the comma after “sick”.

Line 407 and other places: Please check how you mark the start of a new subsection (new line, indent, blank line) and use the same all through the manuscript.

Line 413-415: Please use italics for all citations, and use indent (or not) for all citations.

Line 458: Please remove the comma after “vendors”.

Line 497: Please write out numbers less than twelve in letters.

Line 524: Please remove the comma after “disease restrictions”.

Line 528: Please remove the comma after “lameness and death”.

Line 530: Please revise the sentence (remove the subclausation).

Line 619: Please place the reference at the end of the sentence.

Line 762: Please use the abbreviation (SR) once introduced.

Response: All the above comments highlighted on grammar and typos have been revised.

We look forward to your response and hope the revisions will enable you to accept this version for publication.

Sincerely,

Jane Namatovu, Peter Lule, Marsy Asindu, Zoë A. Campbell, Dan Tumusiime, Henry Kiara, Bernard Bett, Kristina Roesel, Emily Ouma

---

## [Decision Letter · Decision Letter 5]

6 Feb 2025

PONE-D-24-01412R5Gender roles in ruminant disease management in Uganda: Implications for the control of peste des petits ruminants and Rift Valley fever.PLOS ONE

Dear Dr. Jane,

Thank you for submitting your manuscript to PLOS ONE. After careful consideration, we feel that it has merit but does not fully meet PLOS ONE’s publication criteria as it currently stands. Therefore, we invite you to submit a revised version of the manuscript that addresses the points raised during the review process.

**ACADEMIC EDITOR: ** Dear authors, please revise your manuscript carefully, addressing all comments raised by reviewer 3.

We look forward to receiving your revised manuscript.

Kind regards,

Nussieba A. Osman, Dr. Med. Vet.

Academic Editor

PLOS ONE

Journal Requirements:

Reviewers' comments:

Reviewer's Responses to Questions

**Comments to the Author**

1. If the authors have adequately addressed your comments raised in a previous round of review and you feel that this manuscript is now acceptable for publication, you may indicate that here to bypass the “Comments to the Author” section, enter your conflict of interest statement in the “Confidential to Editor” section, and submit your "Accept" recommendation.

Reviewer #3: (No Response)

2. Is the manuscript technically sound, and do the data support the conclusions?

Reviewer #3: Yes

3. Has the statistical analysis been performed appropriately and rigorously? 

Reviewer #3: N/A

4. Have the authors made all data underlying the findings in their manuscript fully available?

Reviewer #3: Yes

5. Is the manuscript presented in an intelligible fashion and written in standard English?

Reviewer #3: Yes

6. Review Comments to the Author

Reviewer #3: Dear authours,

I have only some minor, editorial comments on this version. I do not need to see the paper again, but think that these remaining comments can be dealt with between the authors and the editor.

Minor comments:

Line 55-56, 817: Please put the reference at the end of the sentence.

Line 135, 390: I don’t think the author guidelines includes using an indent to start a new section. Please check and unify all through the manuscript.

Line 162: Please replace the colon at the end of the sentence with a full stop.

Line 244-246 and line 254-256: These two sentences seem to partly convey the same message, please consider merging.

Line 267-268: “by/via/using/over” seems to be missing before “telephone”.

Line 281: The information about Covid-19 is repetition and does not belong in this section.

Line 317: Please remove the space after the “/”.

Line 318: I think “participants” would read better than “their” here, please consider changing.

Line 325: I think that by “quotations” you might mean “quotes”?

Line 369: I presume the number of ticks represent the number of FGDs in which at least one participant mentioned the respective measures? Can you please explain this in the foot note.

Line 405-408: This is reported also in the introduction. Is it a result or background information? Please don’t duplicate.

Line 794: By “socio” do you mean “social”?

Line 826: Please remove “other”.

Line 996: Please consider if the * after E should be removed in the reference list.

7. PLOS authors have the option to publish the peer review history of their article (what does this mean? ). If published, this will include your full peer review and any attached files.

**Do you want your identity to be public for this peer review?** For information about this choice, including consent withdrawal, please see our Privacy Policy .

Reviewer #3: No

---

## [Author Response · Author response to Decision Letter 6]

27 Feb 2025

Dr. Nussieba A. Osman

Academic Editor

PLOS ONE

February 27th, 2025

Re: Revision required for manuscript PONE-D-24-01412R5

Dear Dr. Osman,

We would like to thank the academic editor and the reviewer once again for taking the time to review our manuscript and for the helpful comments. We have addressed the comments raised and corrected some typographical errors.

These changes are detailed below.

Editor’s Comments:

Please review your reference list to ensure that it is complete and correct. If you have cited papers that have been retracted, please include the rationale for doing so in the manuscript text or remove these references and replace them with relevant current references.

Any changes to the reference list should be mentioned in the rebuttal letter that accompanies your revised manuscript. If you need to cite a retracted article, indicate the article’s retracted status in the References list and also include a citation and full reference for the retraction notice.

Response: Thanks very much for these valuable comments. We have rechecked the reference list thoroughly and updated the accession numbers for two references: Overholt et al (1985), reference 14 and Dione et al (2019), reference 74.

Reviewer’s comments:

Comment: Line 281: The information about Covid-19 is repetition and does not belong in this section.

Response: Thank you, the information about Covid-19 has been deleted from the text.

Comment: Line 325: I think that by “quotations” you might mean “quotes”?

Response: Thank you for the suggestion. This has been revised to quotes. Please refer to line 323.

Comment: Line 996: Please consider if the * after E should be removed in the reference list.

Response: Thank you for highlighting this. Reference 23 has been updated accordingly. Please refer to line 990.

Comment: Line 794: By “socio” do you mean “social”?

Response: Yes, this has been revised to social. Please refer to Line 788.

Comment: Line 244-246 and line 254-256: These two sentences seem to partly convey the same message, please consider merging.

Response: Thank you for the suggestion. This has been revised accordingly. Please refer to lines 252-254.

Comment: Line 369: I presume the number of ticks represent the number of FGDs in which at least one participant mentioned the respective measures? Can you please explain this in the foot note.

Response: Thank you. This has been revised to; the number of FGDs in which at least one participant mentioned the respective measure. Please refer to line 367.

Comment: Line 135, 390: I don’t think the author guidelines includes using an indent to start a new section. Please check and unify all through the manuscript.

Response: This has been unified through the manuscript.

Comment: Line 405-408: This is reported also in the introduction. Is it a result or background information? Please don’t duplicate.

Response: Thank you for pointing this out. We have retained this information only in the background. Please refer to lines 88-91.

Typographical errors

Line 55-56, 817: Please put the reference at the end of the sentence.

Line 162: Please replace the colon at the end of the sentence with a full stop.

Line 267-268: “by/via/using/over” seems to be missing before “telephone”.

Line 317: Please remove the space after the “/”.

Line 318: I think “participants” would read better than “their” here, please consider changing.

Line 826: Please remove “other”.

Response: We have revised all the above typographical errors

We look forward to your response and hope the revisions will enable you to accept this version for publication.

Sincerely,

Jane Namatovu, Peter Lule, Marsy Asindu, Zoë A. Campbell, Dan Tumusiime, Henry Kiara, Bernard Bett, Kristina Roesel, Emily Ouma

---

## [Editor Report · Decision Letter 6]

28 Feb 2025

Gender roles in ruminant disease management in Uganda: Implications for the control of peste des petits ruminants and Rift Valley fever.

PONE-D-24-01412R6

Dear Dr. Jane,

We’re pleased to inform you that your manuscript has been judged scientifically suitable for publication and will be formally accepted for publication once it meets all outstanding technical requirements.

Kind regards,

Nussieba A. Osman, Dr. Med. Vet.

Academic Editor

PLOS ONE

---

## [Editor Report · Acceptance letter]

PONE-D-24-01412R6

PLOS ONE

Dear Dr. Jane,

I'm pleased to inform you that your manuscript has been deemed suitable for publication in PLOS ONE. Congratulations! Your manuscript is now being handed over to our production team.

Kind regards,

on behalf of

Dr. Nussieba A. Osman

Academic Editor

PLOS ONE